# Nivolumab plus ipilimumab induce hyper-progression in renal medullary carcinoma: results of a phase II trial and preclinical evidence

Therapeutic options for patients with renal medullary carcinoma (RMC) are limited. Here we report the results of a phase II clinical trial (NCT03274258) of anti-PD1 nivolumab plus anti-CTLA4 ipilimumab in patients with RMC, with objective response rate as primary outcome. Enrollment was halted for futility at a prespecified interim analysis as all 10 treated patients experienced rapid disease progression. 5/10 met radiological criteria for hyperprogression and median progression-free survival (secondary outcome) was 1.38 months (95% confidence interval: 1.28, 1.60). In a post-hoc single-cell RNA sequencing analysis, data from patients with RMC before and after nivolumab plus ipilimumab treatment indicated that immune checkpoint therapy (ICT) triggered an interferon-γ response that induced a "myeloid mimicry" program in tumor cells, regulated by the CEBPB / p300 axis and linked to proliferation and hyperprogression. In preclinical experiments using an immunocompetent somatic mosaic genetically engineered mouse model of RMC, combination ICT accelerated tumor growth while activating myeloid-affiliated transcriptional circuits. Selective pharmacologic inhibition of p300 suppressed this program and restored sensitivity to ICT. These findings reveal an adaptive mechanism of resistance to ICT in RMC and support targeting master myeloid regulators to enable therapeutic benefit.

While currently available therapies have substantially improved the clinical outcomes of many patients with kidney cancers, these treatments are typically ineffective against highly aggressive SMARCB1-deficient kidney malignancies such as renal medullary carcinoma (RMC)[1–4]. Although rare, RMC is the third most common kidney cancer among adolescents and young adults, comprising the vast majority of SMARCB1-deficient renal malignancies, and predominantly afflicting young individuals of African descent who harbor the sickle cell trait[2,5]. RMC is three times more likely to arise from the right kidney compared with the left[5]. Previous studies have demonstrated that increased regional ischemia induced by red blood cell sickling in the medullary

vasa recta of individuals with sickle hemoglobinopathies, particularly in the right kidney, predisposes renal inner medulla cells to SMARCB1 loss, potentially causing RMC[5–7]. The median overall survival of patients diagnosed with RMC is only 13 months, and less than 5% survive beyond 3 years[1,8,9]. Therefore, more effective therapies are desperately needed.

We previously demonstrated that RMC exhibits a "hot" immune-inflamed phenotype characterized by upregulation of immune checkpoints such as cytotoxic T-lymphocyte-associated protein 4 (CTLA-4) and programmed cell death protein 1 (PD-1) in the tumor immune microenvironment[10]. However, whereas one published case

✉e-mail: jgao1@mdanderson.org; lwang22@mdanderson.org; ggenovese@mdanderson.org; pmsaouel@mdanderson.org

report noted a response of RMC to anti-PD-1 immune checkpoint therapy (ICT) with nivolumab[11], two additional case reports suggested poor efficacy of nivolumab alone or in combination with anti-CTLA-4 blockade using ipilimumab[12]. Furthermore, in a phase 2 basket trial of anti-PD-1 monotherapy with pembrolizumab, all five enrolled patients with RMC consistently showed progressive disease, with a median time to progression of only 8.7 weeks, irrespective of PD-L1 expression or tumor-infiltrating lymphocyte (TIL) levels[13].

In this work, we report the results of a prospective clinical trial of nivolumab plus ipilimumab specifically designed for patients with RMC (Supplementary Fig. S1), along with correlative results and pre-clinical experiments to elucidate the mechanisms of RMC resistance to ICT in a transplant model of a syngeneic cell line derived from our immunocompetent somatic mosaic genetically engineered mouse model (SM-GEMM) of RMC[7]. RMC tumor cells undergo distinct transcriptional reconfiguration by engaging interferon gamma receptor 1 (IFNGR1) signaling following combination ICT, leading to the activation of myeloid-affiliated transcriptional circuits mediated by the CEBPB / p300 complex that can promote cell proliferation, resulting in hyperprogression. Inhibition of this adaptive "myeloid mimicry" mechanism by either IFNGR1 knockout or pharmacologic inhibition of p300 can induce antitumor responses to ICT in our immunocompetent SM-GEMM of RMC.

## Results

### Nivolumab plus ipilimumab is associated with hyperprogression in RMC patients

A total of 10 patients with RMC started treatment between June 2018 and December 2020 (Fig. 1a). Table 1 presents the baseline demographic and clinical characteristics. Patients had a median age of 31.5 years (20–77 years) and were primarily identified as black (90%), male (70%), and had an Eastern Cooperative Oncology Group (ECOG) performance status of 1 (60%). With only one patient showing an unconfirmed partial response (PR) on a single measurement (Fig. 1b), the trial was halted for futility after enrolling 10 patients. All enrolled patients experienced rapid progression both clinically and radiologically (Fig. 1c, d) with 5 out of the 10 patients meeting established radiological criteria for hyperprogression[14] (Fig. 1b–d). The median number of nivolumab plus ipilimumab infusions were 2 (range 1–6), with 7 of 10 patients receiving 2 infusions, and one patient received 1, 4, and 6 infusions. The median progression-free survival (PFS) was only 1.38 months (95% CI 1.28 – 1.60; Fig. 1e) and median overall survival (OS) was 8.23 months (95% CI 3.45 – NE; Fig. 1f). Additional pre-specified secondary endpoints such as time to objective response rate (ORR), duration of response, and disease control rate (DCR) were not estimated due to the lack of confirmed objective responses and rapid progression in all 10 patients enrolled.

Supplementary Table S1 presents treatment emergent adverse events (AEs) at least possibly related to study treatment. No unexpected toxicity signals were observed. Hypotension in one patient was the only related grade 4 event. This qualified as the only pre-specified extreme toxicity (TOX) event per trial protocol (the complete trial protocol is provided in the Supplementary note). Related grade 3 events included increased alanine aminotransferase, anemia, and pain ($n = 1$ each). Overall, the most commonly related AE was decreased neutrophil count ($n = 3$). Supplementary Table S2 lists all treatment emergent AEs regardless of attribution. All 10 patients had at least one AE. Grade 3 or higher events included pain ($n = 2$), and abdominal pain, increased alanine aminotransferase increased, anemia, hypotension, and pericardial tamponade ($n = 1$ each). The most common AEs overall were back pain ($n = 4$), pain ($n = 3$), constipation ($n = 3$), dyspnea ($n = 3$), neutrophil count decreased ($n = 3$), and proteinuria ($n = 3$).

### Myeloid-like transcriptional features emerge in RMC liver metastases after treatment with nivolumab plus ipilimumab

To elucidate the mechanisms behind the observed hyperprogression, we first performed post-hoc single-cell RNA sequencing (scRNA-seq) on immunotherapy-naïve fresh tumor biopsy tissues from 7 patients with RMC, as well as on tissues collected immediately following progression on nivolumab plus ipilimumab from 2 patients with RMC (Fig. 2a and Supplementary Tables S3, S4). Utilizing scRNA-seq, we captured transcriptomes of 23,880 cells after stringent quality control. These cells were broadly categorized into 10 major cell types based on the expression of canonical marker genes (Fig. 2b and Supplementary Figs. S2, S3). Among 9 patients, tumor cells were identified in 7 patients and included for further analysis (Fig. 2c). The two patients without identifiable tumor cells were omitted from further analysis (Supplementary Tables S3, S4). Tumor cells were identified within clusters that harbored extensive copy-number variations (CNVs) across their genomes, as inferred from scRNA-seq data (Supplementary Fig. S4)[15]. Pathway enrichment analysis revealed the anticipated upregulation of inflammatory pathways, including interferon gamma (IFNγ), in the T cell compartment following treatment with nivolumab plus ipilimumab (Fig. 2d). Notably, significant enrichment of pathways associated with DNA replication and cell cycle pathways was also observed in the tumor cell compartment, consistent with the clinically observed hyperprogression (Fig. 2d).

Analysis of the top 100 differentially expressed genes in tumor cells treated with the combination of nivolumab and ipilimumab showed the enrichment of a myeloid marker, *S100A9*, encoding S100 calcium-binding protein A9 (also known as myeloid-related protein 14 or MRP14), as one of the top-scoring hits (Supplementary Fig. S5). Furthermore, the proportion of RMC tumor cells expressing the myeloid lineage marker *S100A9* and the cell proliferation marker *MKI67* was significantly higher following ICT treatment (Fig. 2e). S100A9 is a myeloid lineage marker that has been reported to positively regulate expression of CCAAT/enhancer-binding protein beta (CEBPB)[16], which is also highly expressed in cells committed to the myelomonocytic hematopoietic lineage, where it cooperates with the co-activator p300, encoded by *EP300*, to activate expression of myeloid-specific genes[17]. Indeed, *CEBPB* and *EP300*, along with genes related to the Mitogen-Activated Protein Kinase (MAPK) signaling cascade such as *MAP3K1*, *MAP2K2*, and *MAP3K11* were significantly upregulated in RMC tumor cells following treatment with nivolumab plus ipilimumab, (Fig. 2f). Paradoxically, although there was a decrease in canonical interferon pathways in RMC tumor cells after treatment (Fig. 2d), these same tumor cells expressed significantly higher levels of *IFNGR1*, which encodes the interferon gamma receptor 1 (Fig. 2f), a receptor that is highly expressed on macrophages and monocytes and promotes antigen presentation on non-immune cells, as well as *JAK1*, a downstream effector of interferon gamma signaling[18,19]. A sensitivity analysis using only the subset of patients with liver biopsies showed similar findings (Supplementary Figs. S6–S8). Notably, while the *EP300* upregulation observed in the full cohort was attenuated under this restriction, its partner gene *CEBPB* retained a robust and highly significant increase (Supplementary Fig. S8), supporting the biological relevance of the EP300–CEBPB axis even in this more stringent setting.

We subsequently interrogated longitudinally collected blood and tissue samples from our prospective clinical trial. First, we performed mass cytometry by time-of-flight (CyTOF) on peripheral blood mononuclear cell (PBMC) collected at baseline and following nivolumab plus ipilimumab (Supplementary Figs. S9–S13). A significant increase in circulating total CD4 + as well as CD4 + EM1 + T cells was noted following nivolumab plus ipilimumab therapy (Supplementary Fig. S13). CD4 + EM1 + T cells are known to home to tissues and produce IFN-γ within hours of stimulation[20]. Multiplex immunofluorescence of these tumor tissues showed an increase in CD8 + T cells migrating from the stroma to the panCK+ tumor cell nests following ICT treatment, along

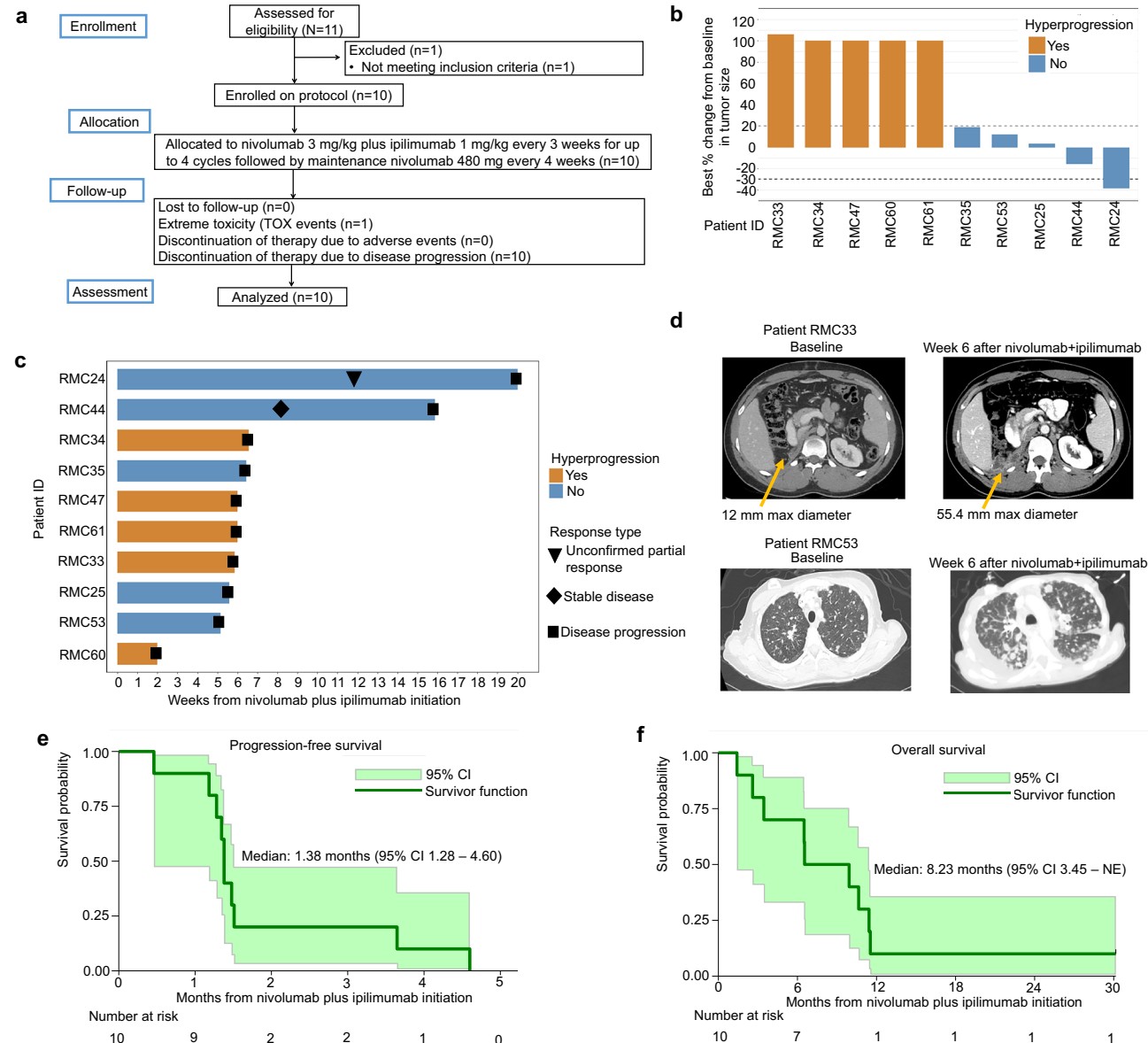

**Fig. 1 | Hyperprogression of RMC in response to immune checkpoint therapy (ICT) with nivolumab and ipililumab. a** CONSORT diagram. **b** Waterfall plot showing the maximum change in the sum of the longest dimensions in each of the 10 evaluable patients treated with nivolumab + ipilimumab. Hyperprogression was defined as progressive disease in the first 8 weeks after treatment initiation, a 10 mm minimum increase in the measurable lesions, plus: (i) an increase of ≥ 40% in the sum of target lesions compared with baseline and/or (ii) an increase of ≥ 20% in the sum of the target lesions compared with baseline plus the appearance of new lesions in at least two different organs[14]. **c** Swimmer plot showing the duration (in weeks) that each of the 10 patients enrolled on the trial remained on therapy. **d** Representative images at baseline and following nivolumab plus ipilimumab. A 4.6-fold increase was noted in the measurable target lesion shown in patient RMC33 (top). Aggressive appearance of multiple new non-target lesions was noted in the lungs within 6 weeks from treatment initiation in patient RMC53 (bottom). **e, f** Progression-free survival (**e**) and overall survival (**f**) from treatment initiation with nivolumab plus ipilimumab. The solid line is the Kaplan–Meier estimate as the measure of center. The shaded light green areas represent the 95% confidence bands for each curve. Source data are provided as a Source Data file.

with an increase in IFNγ in the tumor microenvironment (TME) and a concomitant enrichment in tumor cells expressing *S100A9* and *IFNGR1* (Fig. 3a–d and Supplementary Fig. S14). Bulk RNA-sequencing of the longitudinally collected tumor tissues again revealed enrichment for genes associated with IFNγ signaling, DNA replication and cell cycle pathways, corroborating our scRNA-seq analysis (Supplementary Fig. S15 and Supplementary Table S6)

## RMC liver metastases have increased transcriptional similarity to myeloid cells after treatment with nivolumab plus ipilimumab

To measure whether tumor cells treated with ICT are undergoing cell state changes, we utilized the single-cell entropy (SCENT)[21] algorithm

to quantify tumor cell entropy, a robust proxy for a cell's differentiation potential. Based on this analysis, tumor cells demonstrated a significantly higher signaling entropy ratio post nivolumab plus ipilimumab (Fig. 3e), consistent with an increased complexity and a broader range of functional states, suggesting that tumor cells display a higher state of cellular plasticity upon treatment (Fig. 3f)[21]. Furthermore, to robustly quantify the transcriptomic similarity between tumor cells and myeloid cells, we calculated at single cell resolution multiple metrics, including Manhattan and Euclidean distance metrics, the Cosine and Pearson correlation-based directional similarity metrics, as well as the Phi and Bray-Curtis proportionality and compositional dissimilarity metrics[22,23]. To eliminate potential tissue site biases, we limited our analyses to the subset of patients with liver

**Table 1 | Baseline demographics, clinical and prior treatment characteristics**

| | Total (*N* = 10) |
|---|---|
| **Baseline Characteristics** | |
| **Age, Median (Range)** | 31.5 (20.0, 77.0) |
| **Sex**, n (%) | |
| Female | 3 (30.0%) |
| Male | 7 (70.0%) |
| **Ethnicity**, n (%) | |
| Black or African American | 8 (80.0%) |
| Hispanic or Latino | 1 (10.0%) |
| White or Caucasian | 1 (10.0%) |
| **Sickle hemoglobinopathy**, n (%) | |
| Sickle cell trait | 9 (90%) |
| None | 1 (10%) |
| **RMC laterality**, n (%) | |
| Right kidney | 8 (80%) |
| Left kidney | 2 (20%) |
| **ECOG performance Status**, n (%) | |
| 0 | 3 (30.0%) |
| 1 | 6 (60.0%) |
| 2 | 1 (10.0%) |
| **Stage at initial diagnosis of RMC**, n (%) | |
| III | 2 (20%) |
| IV | 8 (80%) |
| **Prior cytoreductive nephrectomy**, n (%) | |
| Yes | 6 (60%) |
| No | 4 (40%) |
| **Prior platinum-based chemotherapy**, n (%) | |
| Yes | 5 (50%) |
| No | 5 (50%) |
| **Number of prior systemic treatments, Median (Range)** | 1 (0, 3) |

metastases biopsies (Supplementary Table S3 and Supplementary Fig. S16). All four dissimilarity metrics (Euclidean, Manhattan, Bray-Curtis, and Phi) consistently indicated closer similarity between tumor cells and myeloid cells in PostNI compared to baseline (Fig. 3g, h). Likewise, both similarity metrics (Pearson and Cosine) also showed higher directional similarity in the PostNI group than in the baseline (Fig. 3g, h). These results show that the tumor cells post-treatment demonstrated a significantly greater transcriptomic similarity to myeloid cells than those at baseline. A sensitivity analysis on all seven patient samples from various tissue biopsy sites yielded similar findings (Supplementary Figs. S17, S18).

## Myeloid cells and tumor cells interactions increased after treatment with combination ICT

To assess TME changes, we utilized CellChat[24] to investigate cell–cell communication within the tumor microenvironment, focusing on liver metastases. We compared the interactions between tumor cells and different immune cells, including CD8T, CD4T, NK, myeloid cells and B cells, as shown in Supplementary Fig. S19. The CellChat analysis showed an increase in tumor and myeloid cell interactions as well as a decrease in CD8 T cell and tumor cells interactions in the postNI liver metastases compared with baseline (Supplementary Fig. S19a–c). Similarly, the chord plot in Supplementary Fig. S19d also showed an increase in myeloid interactions with malignant cells in the PostNI tumors. Deeper characterization of the cell types (Supplementary Fig. S19e) revealed enrichment in the postNI tumors of SPP1 + tumor-associated macrophages (TAMs), which can drive immunotherapy

resistance by dysregulating CD8 + T cells[25], and remodel the TME to support tumor cell aggressiveness correlating with worse prognosis[26].

In summary, these results suggest that the IFNγ signaling pathway, predominantly activated by CD8 and CD4 T cells stimulated by ICT (Fig. 2d), is engaging in a non-canonical, 'myeloid mimicry' pathway that is conferring transcriptional reprogramming, adaptation and survival to therapy-induced inflammatory stress. We also showed that myeloid cells, in particular TAMs, and tumor cell interactions increased after combination immunotherapy (Supplementary Fig. S19), further supporting the notion that tumor cells are mimicking the surrounding myeloid cells to adapt to a changing TME. However, the lack of matched baseline and post-treatment samples for single-cell transcriptomics precluded definitive conclusions. We therefore proceeded to investigate whether these findings could be recapitulated in an RMC mouse model.

## Mouse model of RMC recapitulates hyperprogressive disease response to ICT treatment

To confirm hyperprogression and myeloid mimicry in controlled experimental settings, we performed preclinical experiments in an immunocompetent mouse model of RMC (Fig. 4a) that recapitulates the metastatic pattern and aggressive behavior of this disease[7]. We administered standard established in vivo doses[27,28] of anti-mouse PD-1 in combination with anti-CTLA-4 (Fig. 4b). Consistent with the clinical observations of hyperprogression in response to nivolumab plus ipilimumab, the preclinical combination of anti-PD-1 plus anti-CTLA-4 (ICT) resulted in significantly increased primary tumor mass (Fig. 4c–e), as well as a greater metastatic tumor burden compared with IgG control, particularly in the lungs and liver (Fig. 4f, j). Immunohistochemistry analysis of primary tumors confirmed significantly increased mitosis, Ki-67, and MAPK pathway engagement following the combination ICT compared with IgG control in primary kidney tumors (Fig. 5a–g). Tumor cells in our RMC mouse models also express green fluorescent protein (GFP)[7], which allows us to distinguish them from non-malignant mouse cells (Fig. 4a). Multiplex immunofluorescence accordingly revealed that primary tumor tissues from mice treated with combination ICT showed a significant increase in tumor cells co-expressing GFP (marker of tumor cells) and myeloid markers such as S100A9 (Fig. 5f, g), CD68 (Supplementary Fig. S20), and F4/80 (Supplementary Fig. S21). These results confirm that, consistent clinical observations in human specimens, mouse RMC tumor cells upregulate myeloid lineage markers and undergo hyperprogression in response to ICT.

## Genetic knockout of interferon gamma receptor 1 (IFNGR1) sensitizes RMC tumor cells to checkpoint therapy in vivo

Several studies have demonstrated the duality role of interferon signaling as both a key mediator of both tumor cell killing and a contributor to resistance to ICT[29,30]. Type II interferon signaling, in particular, appears to be capable of promoting tumor cell growth and expansion. For example, a recent study noted that IFNγ produced by T cells promoted hyperprogression in lung cancer[31]. Our scRNA-seq analysis of patient tumors confirmed that IFNγ was primarily produced by T cells following ICT (Fig. 2d). In parallel, RMC tumor cells upregulated expression of *IFNGR1* (Fig. 2f), which normally engages IFNγ to activate proliferation of myeloid lineage immune cells via the MAPK signaling pathway as part of the innate immune response[18,19]. Therefore, the upregulation of *IFNGR1* on tumor cells prompted us to further investigate whether this pathway mediates RMC tumor cell proliferation and hyperprogression in RMC in response to ICT.

To assess the direct impact of *IFNGR1* in RMC tumor cells, we used CRISPR gene editing to genetically knockout *IFNGR1* in MSRT1 mouse cell lines (Fig. 6a). Protein analysis following IFNγ treatment of parental MSRT1 cells (IFNGR1^WT) compared to MSRT1 cells with IFNGR1 knockout (IFNGR1^KO) showed that MEK/ERK signaling was markedly

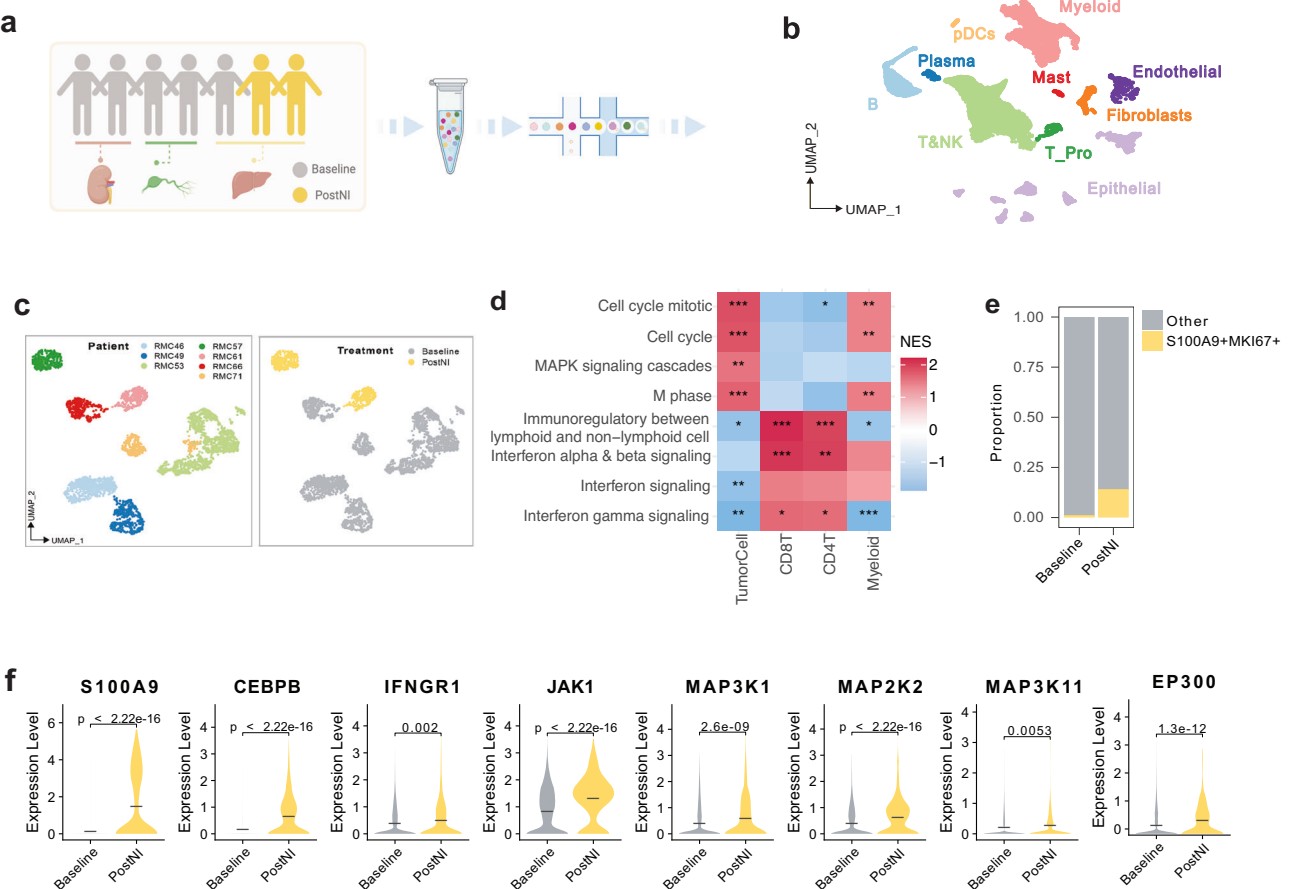

**Fig. 2 | Single cell RNA sequencing (scRNA-seq) of tumor samples from patients with RMC treated with nivolumab (N) and ipilimumab (I). a** Schematic of the scRNA-seq tumor sample collection from different organ sites of patients with RMC at baseline ($n = 7$) and following immune checkpoint therapy (ICT) with N + I ($n = 2$). Created in BioRender. Yan, X. (2025) https://BioRender.com/vqkcx1g. **b** UMAP shows unsupervised clustering analysis of 23,880 single cells. Each dot of the UMAP represents a single cell. Cells are color-coded for their associated cell types. **c** UMAP shows unsupervised clustering analysis of malignant cells. Cells are colored with patient id (left) and treatment condition (right). **d** Comparison of pathway enrichment between baseline and PostNI across different cell types. The heatmap displays normalized enrichment scores for selected pathways. The *p*-values were calculated using an empirical phenotype-based permutation test built into GSEA. Significance is indicated with asterisks (* for *p*-value ≤ 0.05, ** for *p*-value ≤ 0.01, *** for *p*-value ≤ 0.001). **e** Stacked barplot illustrates an increased proportion of *S100A9 + MKI67 +* RMC tumor cells following ICT with nivolumab and ipilimumab (PostNI). **f** Violin plots displaying the increased expression of immune-related genes in PostNI compared to baseline tumors, with the mean shown as the center line. The *p*-value was calculated using the two-sided Wilcoxon rank-sum test. Source data are provided as a Source Data file.

decreased in IFNGR1^KO tumor cells (Fig. 6b), supporting a positive role of IFNGR1 in modulating MAPK pathway activity in response to IFNγ stimulation. To interrogate the effect of interferon signaling on driving myeloid mimicry, we first treated MSRT1 cells with IFNγ in vitro and noted a significant upregulation of S100A9 and mouse myeloid lineage markers, including F4/80, Ly6c, and CD11c (Supplementary Figs. S22–S24). Subsequently, we focused on F4/80, a gold-standard myeloid-specific marker in murine systems[32,33]. Treatment of MSRT1 cells with increasing IFNγ concentrations in vitro resulted in a significant increase in F4/80 expression (Fig. 6c, e and Supplementary Fig. S23). In contrast, treatment with IFNγ did not alter F4/80 levels in IFNGR1^KO MSRT1 cells (Fig. 6d, f).

Transplantation of IFNGR1^KO cells into the right kidneys of immunocompetent C57BL/6 J pure background mice with sickle cell trait showed significantly decreased growth compared to IFNGR1^WT cells treated with both control (IgG) and ICT (anti-CTLA-4 and anti-PD-1) (Fig. 7a–d). Remarkably, IFNGR1^KO tumors treated with ICT had complete remission in 5 out of 6 mice after 90 days (Fig. 7b, d) with no evidence of tumor formation by MRI, demonstrating that genetic knockout of IFNGR1 selectively sensitized RMC tumor cells to immunotherapy. Notably, tumors with IFNGR1^KO treated with control

IgG were significantly larger compared to IFNGR1^WT treated with control IgG, suggesting a pleiotropic role of IFNGR1 in tumor growth (Fig. 7d). Furthermore, genetic knockout of IFNGR1 caused the downregulation of p300 protein (Fig. 7c), suggesting that IFNγ signaling via IFNGR1 is directly mediating p300 activity. In addition, CREB-binding protein (CBP; also known as KAT3A), the paralog of p300 encoded by the *CREBBP* gene, was not significantly upregulated in RMC tumor cells following ICT (log2FoldChange 0.13, *P*-value > 0.5), suggesting that p300 is specifically engaged during RMC hyperprogression. In summary, these results further support the hypothesis that IFNγ signaling can promote RMC tumor cell state changes, prompting further investigation into the pharmacological inhibition of myeloid-affiliated mediator, p300, which is downstream of IFNGR1.

## Pharmacological inhibition of p300 sensitizes RMC to immunotherapy

Our analysis showed that *EP300*, which encodes the histone acetyl-transferase p300, was significantly upregulated on RMC tumor cells after treatment with ICT (Fig. 2f). We also demonstrated that genetic ablation of IFNGR1 caused the loss of p300 (Fig. 7c) and of global

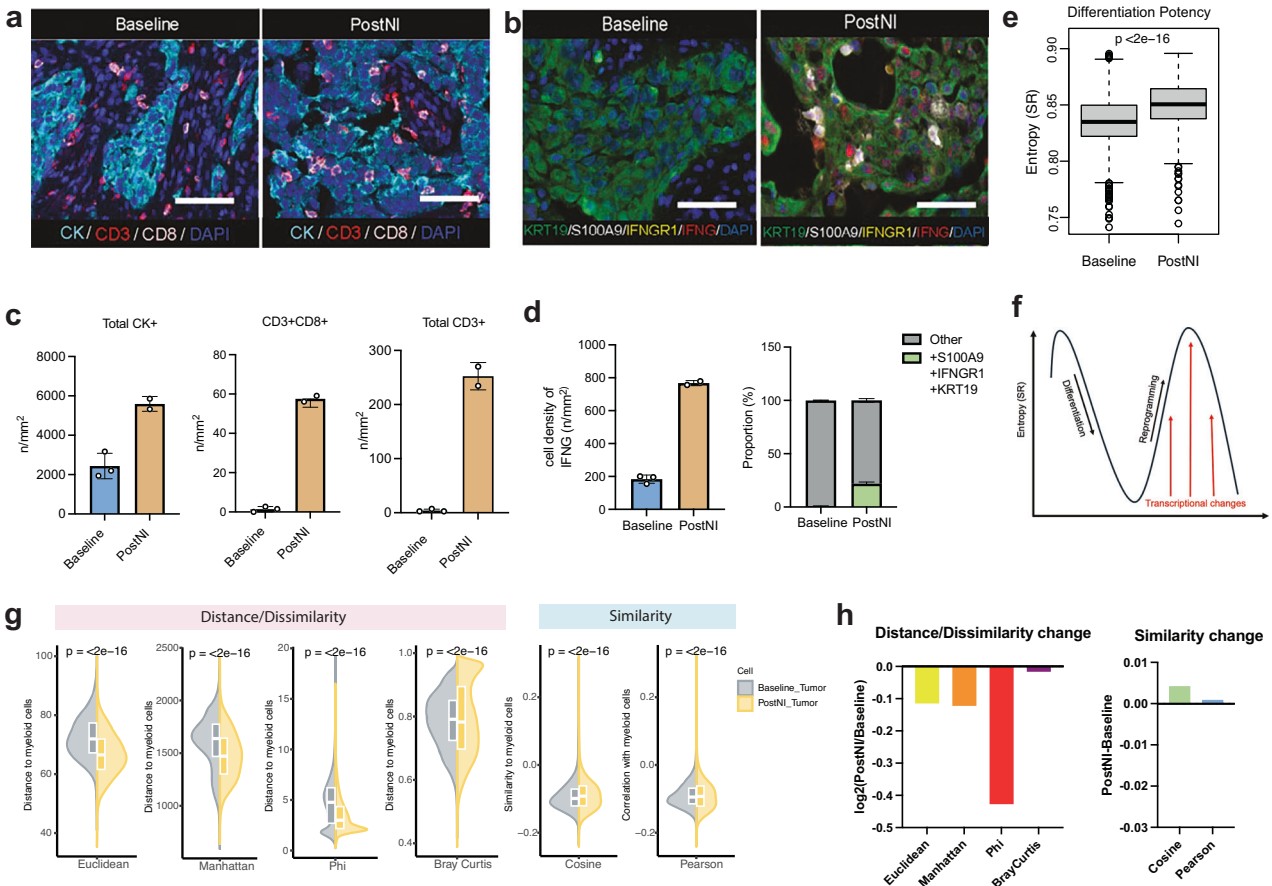

**Fig. 3 | Multiplex immunofluorescence and tumor cell entropy of RMC treated with nivolumab and ipilimumab. a**, **b** Representative images of multiplex immunofluorescence (mIF) of patients at baseline, (*n* = 3) compared to patients treated with nivolumab plus ipilimumab (PostNI, *n* = 2). For (**a**), cytokeratin (panCK, light blue), cluster of differentiation 3 (CD3, red), cluster of differentiation 8 (CD8, pink), and Diamidino-2-phenylindole (DAPI, dark blue). For (**b**), keratin 19 (KRT19, green), S100 calcium-binding protein A9 (S100A9, white), interferon gamma receptor 1 (IFNGR1, yellow), interferon gamma (IFNG, red), and Diamidino-2-phenylindole (DAPI, dark blue). All images are 40X magnification (scale bar 100 μm). **c** Quantification of Fig. 3a protein markers. **d** (Left) Quantification of IFNγ and co-localization of S100A9, IFNGR1, and KRT19, (Right) parts of whole of the percentage of tumor cells co-expressing S100A9, IFNGR1, and KRT19. For (**c**, **d**), all analysis compared patients at baseline (*n* = 3) versus PostNI (*n* = 2). InForm 2.8.2 image analysis software (Akoya Biosciences) was used to quantify images, and the data are expressed as mean value ± SD, and all samples representing biological replicates from different patients. **e** Boxplot shows the tumor cell entropy estimated by signaling entropy rate (SR) at baseline (*n* = 5) and PostNI (*n* = 2). Box-and-whisker plots show all values with range (whiskers), interquartile range (box) and mean (center line). The *p*-value was calculated using the two-sided Wilcoxon rank-sum test. **f** Schematic of the relationship between SR and tumor cell plasticity. Increased tumor cell differentiation potential is associated with higher SR. **g** Violin plot shows the distance/dissimilarity and similarity between liver tumor cells and myeloid cells at baseline (*n* = 1) and PostNI (*n* = 2). Box-and-whisker plots show all values with range (whiskers), interquartile range (box) and mean (center line). The *p*-value was calculated using the two-sided Wilcoxon rank-sum test. **h** Bar plot shows the log fold change of the distance/dissimilarity (including Euclidean distance, Manhattan distance, Phi and Bray Curtis) and the change of similarity (including cosine and Pearson correlation) between liver tumor cells and myeloid cells at baseline (*n* = 1) and PostNI (*n* = 2). Source data are provided as a Source Data file.

acetylation of Lys-27 residue of histone 3 (H3K27Ac), a histone mark associated with p300 activity (Fig. 6b). *EP300* which is activated by ERK signaling[34], plays a critical role in myeloid differentiation[35], and is responsible for gene activation via acetylation of Lys-27 residue of H3K27 acetylation[36]. Therefore, inhibition of EP300 has promising potential to inhibit myeloid mimicry in RMC tumor cells. To pharmacologically inhibit the acetyltransferase activity of p300, we developed IACS-16898[37], an orally available inhibitor targeting the bromodomains of p300/CBP, and tested whether pharmacological targeting of this downstream pathway could restore sensitivity to ICT in our immunocompetent mouse model of RMC. A structure-based design approach was undertaken to develop IACS-16898 (Fig. 8a). An in vitro AlphaScreen technology binding assay showed that IACS-16898 is a potent inhibitor of CREB-binding protein (CBP), the paralog of p300, with an IC$_{50}$ of 3 nM versus a reported IC$_{50}$ of 4790 nM for BRD4 (Fig. 8b and Supplementary Fig. S25). The selectivity of IACS-16898 against the bromodomain of CBP and p300 was further evaluated using a

DiscoverX BROMOscan assay, which showed high affinity for CBP and p300, demonstrating K$_d$ of 0.23 and 0.27 nM, respectively, and a higher than 300-fold (mostly higher than 1000-fold) selectivity against other bromodomains (Fig. 8b and Supplementary Tables S8, S9).

In vitro and in vivo characterization of IACS-16898 pharmacologic activity was performed in the DOHH2 non-Hodgkin lymphoma cell line, previously established to be highly sensitive to CBP/p300 inhibition[38,39]. Treatment of DOHH2 cells with IACS-16898 reduced global H3K27 acetylation after 24 h (Supplementary Fig. S26) and suppressed cell proliferation at nanomolar concentrations with an IC$_{50}$ of 29 ± 9 nM (Supplementary Fig. S27). The in vivo preclinical pharmacokinetics of IACS-16898 in mice, rats, dogs, and monkeys are shown in Supplementary Tables S10, S11. IACS-16898 showed an AUC of 802 hr * μM (Supplementary Table S10) and low in vivo clearance of 4 mL/min/kg, with a half-life of 4 h in mice (Supplementary Table S11), making it suitable for in vivo therapeutic use in mice. Subsequently, in vivo treatment of subcutaneous DOHH2 xenografts with IACS-16898

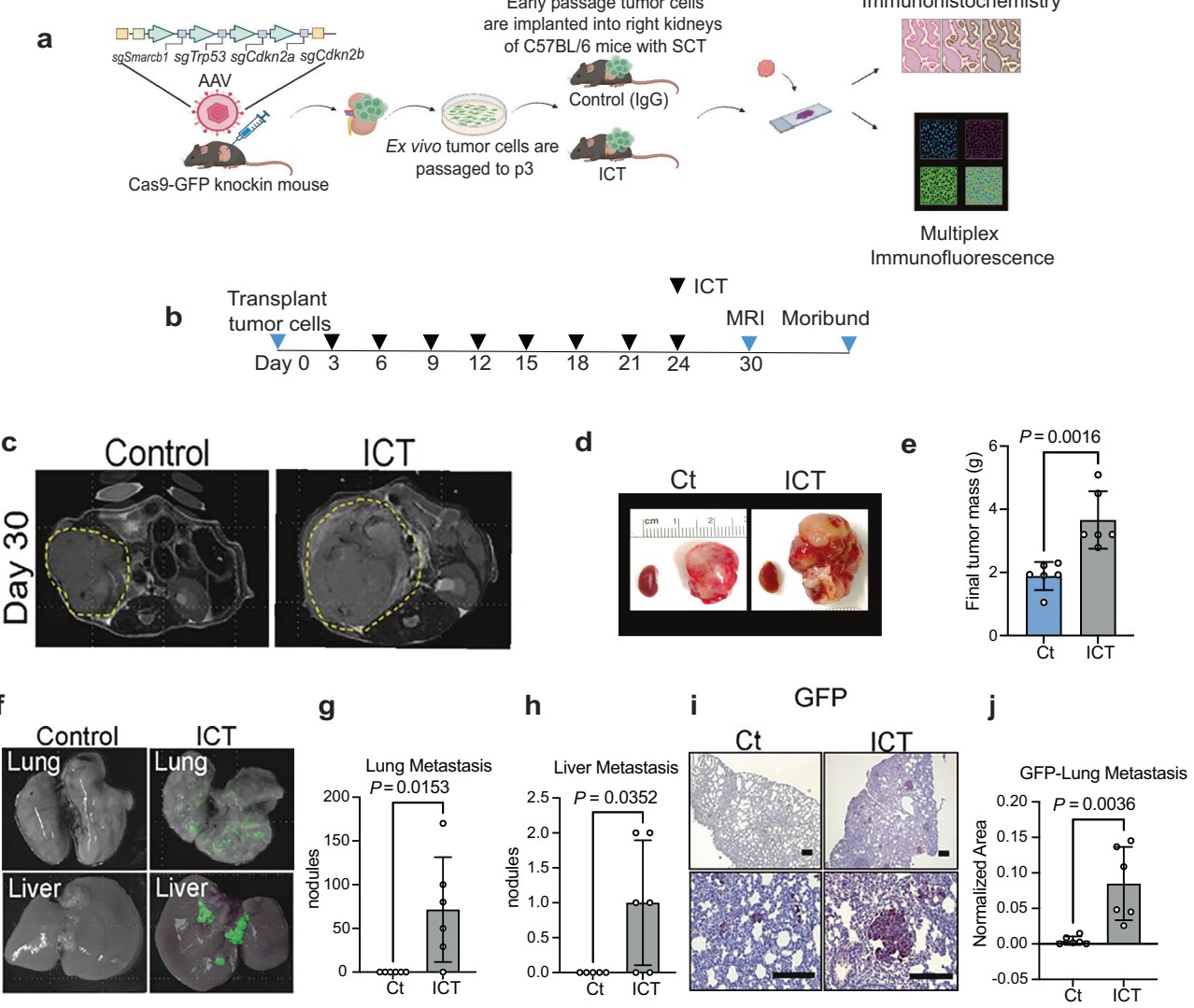

**Fig. 4 | Hyperprogression in response to combination immune checkpoint therapy (ICT) in genetically engineered mouse model of RMC. (a)** Schema showing the workflow of the in vivo preclinical study using the ex vivo RMC tumor cell line (MSRT1) generated from a mouse model of RMC in a C57BL/6 strain[7]. Created in BioRender. Soeung, M. (2025) https://BioRender.com/g4wcktf. **b** Schema of dosing regimen for the in vivo preclinical ICT study. **c** Representative magnetic resonance imaging (MRI) of mice bearing right kidney tumors on day 30 of treatment. Tumor boundaries are traced in the dotted yellow line to illustrate the difference in the size of the control- versus the ICT- treated RMC tumors. **d** Gross representative images of final primary right kidney tumors from mice treated with IgG control (Ct) and anti-PD-1 plus anti-CTLA-4 (ICT). **e** Quantification of final primary tumors from the right kidneys of mice treated with either Ct or ICT. Each dot represents one mouse for a total of $n = 6$ mice per treatment group.

**f** Representative gross images of liver and lung metastasis from mice treated with Ct or ICT. Tumor cells stably express green fluorescent reporter (GFP). The GFP in the image reveals metastatic implants in the liver and lung. **g** Quantification of metastatic nodules in the liver based on GFP in $n = 6$ mice per group. **h** Quantification of metastasis nodules in the lungs based on GFP in $n = 6$ mice per group. **i** Representative images at 4 x (top panels) and 40 x (bottom panels) of immunohistochemistry (IHC) staining of GFP in the lungs of mice treated with Ct or ICT. All slides were counterstained with hematoxylin. **j** Quantification of immuno-histochemistry staining of the total area of GFP in the lungs of mice treated with Ct ($n = 6$) or ICT ($n = 6$). Data in (**e**), (**g**), (**h**) and (**j**) are expressed as mean value ± SD, with $P$-value calculated by two-sided student's $t$ test, and samples are biological replicates representing different mice in the study. Source data are provided as a Source Data file.

led to significantly lower tumor volumes compared with vehicle control (Supplementary Fig. S28), and treatment was well tolerated with no more than 5–10% reduction in animal body weight compared with vehicle control throughout the treatment (Supplementary Fig. S29). To determine the in vivo target engagement by IACS-16898, we measured the levels of B-cell lymphoma 6 (BCL6), an established lymphoma oncogenic driver amplified in DOHH2 cells[40]. Significant downregulation of BCL6 transcript levels by 95% ($p = 0.0001$) were noted in DOHH2 tumors harvested 16 h after last dosing of IACS-16898 at 50 mg/Kg compared with vehicle control (Supplementary Fig. S30).

We subsequently used IACS-16898 to determine whether pharmacological targeting of p300 could restore sensitivity to ICT in our immunocompetent mouse model of RMC (Fig. 8c–f). Successful in vivo inhibition of CBP/p300 activity was demonstrated by a reduction in H3K27 acetylation following IACS-16898 treatment (Fig. 8d). Whereas IACS-16898 monotherapy did not yield any antitumor responses and anti-PD-1 plus anti-CTLA-4 (ICT) alone induced hyperprogression, the combination of IACS-16898 with ICT significantly decreased RMC tumor growth (Fig. 8e, f). Immunofluorescent analysis of ex vivo tumors after 30 days of treatment showed a significant decrease in F4/80 expression (Fig. 8g, h) and S100A9 expression (Supplementary Figs. S31, S32) in tumors treated with the combination of ICT and IACS-16898. Overall, our findings suggest that the hyperprogression of RMC in response to ICT occurs via engagement of

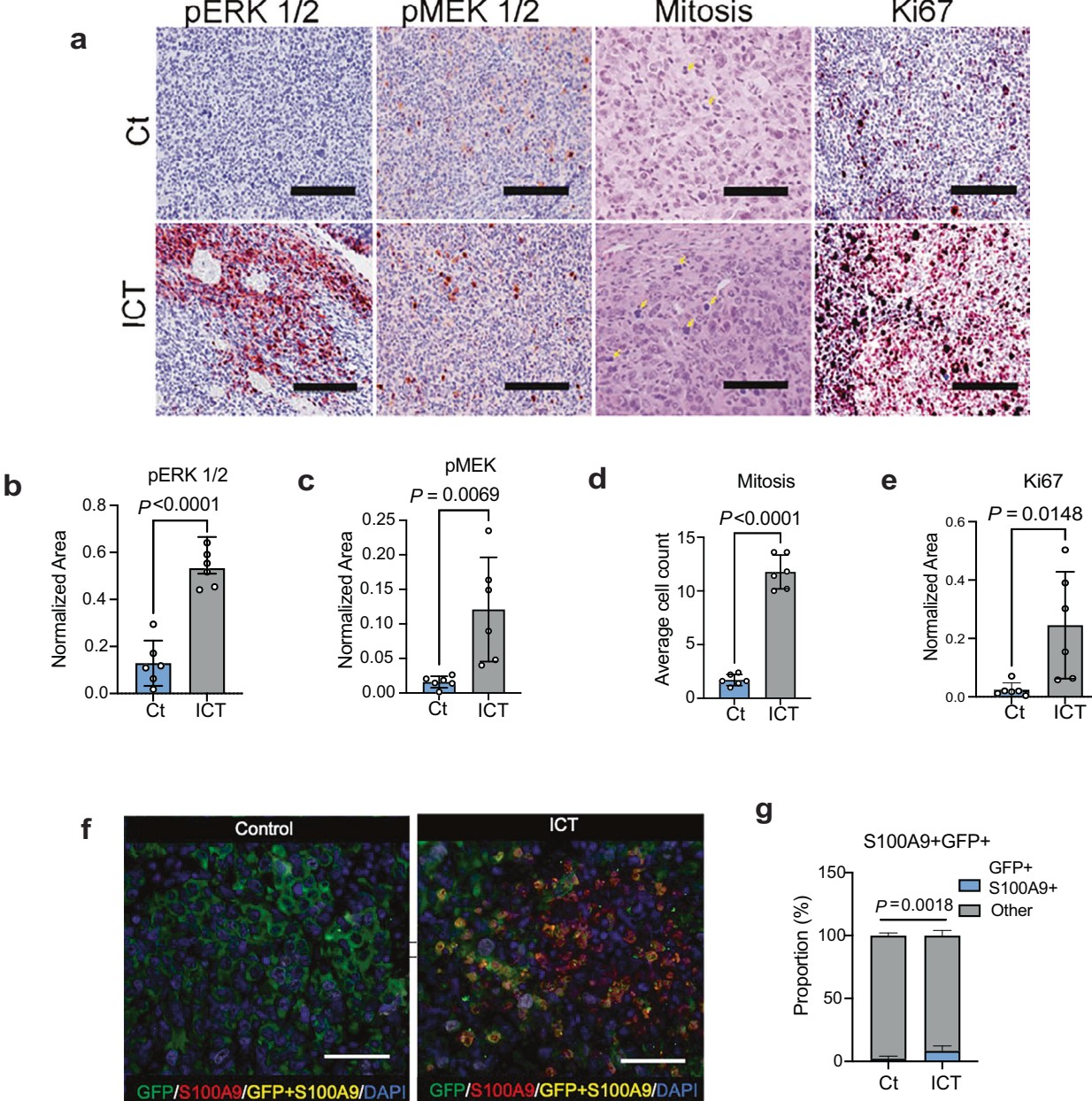

**Fig. 5 | Effect of immune checkpoint therapy on MAPK signaling, proliferation, and myeloid mimicry in primary RMC mouse tumors. a** IHC staining of mitogen-activated protein kinase (MAPK) signaling pathway and cell cycle markers in primary RMC kidney tumors of mice treated with IgG control (Ct) or anti-PD-1 plus anti-CTLA-4 (ICT). All samples were counterstained with hematoxylin to account for the total number of cells within each image. All images are 40X magnification, and the width of the scale bar is 100 μm. **b−e** Quantifications of IHC analysis of phospho-protein kinase R (PKR)-like endoplasmic reticulum kinase 1/2 (pERK 1/2), (**c**) phospho-MAPK/ERK kinase 1/2 (pMEK 1/2), (**d**) mitosis events in hematoxylin and eosin (H&E) staining, and (**e**) Ki67 in primary right kidney tumors of mice

treated with Ct ($n = 6$) or ICT ($n = 6$). **f** Representative images of multiplex immunofluorescence analysis comparing RMC mouse models treated with Ct ($n = 6$) compared to mice treated with ICT ($n = 6$). Tumor cells are labeled with GFP. All images are 40X magnification, and the width of the scale bar is 100 μm. **g** Parts of whole of the percentage of tumor cells with GFP co-expressing S100A9 in RMC mouse models treated with Ct ($n = 6$) compared to mice treated with ICT ($n = 6$). Data in (**b−e**) and (**g**) are expressed as mean value ± SD, with $P$ value calculated by two-sided student's $t$ test, and samples are biological replicates representing different mice in the study. All images are 40X magnification. Source data are provided as a Source Data file.

p300, which acts downstream of IFNGR1 and can be prevented by pharmacologic inhibition of p300 prevents this adaptive mechanism, thus restoring sensitivity to ICT (Fig. 8i).

## Discussion

Myeloid mimicry induced by IFNγ signaling was recently described as a mechanism of immune evasion in mouse models of glioblastoma multiforme[41]. However, our findings in both human tissues and mouse models suggest that myeloid mimicry in RMC promotes not only

immune resistance but also hyperprogression. We also noted that RMC tumor cells treated with ICT overexpress IFNGR1 and that genetic knockout of IFNGR1 prevented hyperprogression and sensitized tumors to ICT in pre-clinical settings. Activation of IFNGR1 by IFNγ can induce global H3K27 acetylation in myeloid cells[42]. Similarly, we found that IFNGR1 knockout reduced p300 in RMC cells in the presence of IFNγ. H3K27 acetylation in myeloid cells is typically performed by p300 and its paralog, CBP, encoded by the *CREBBP* gene[35,36,43]. However, only *EP300* and not *CREBBP* was significantly upregulated in RMC

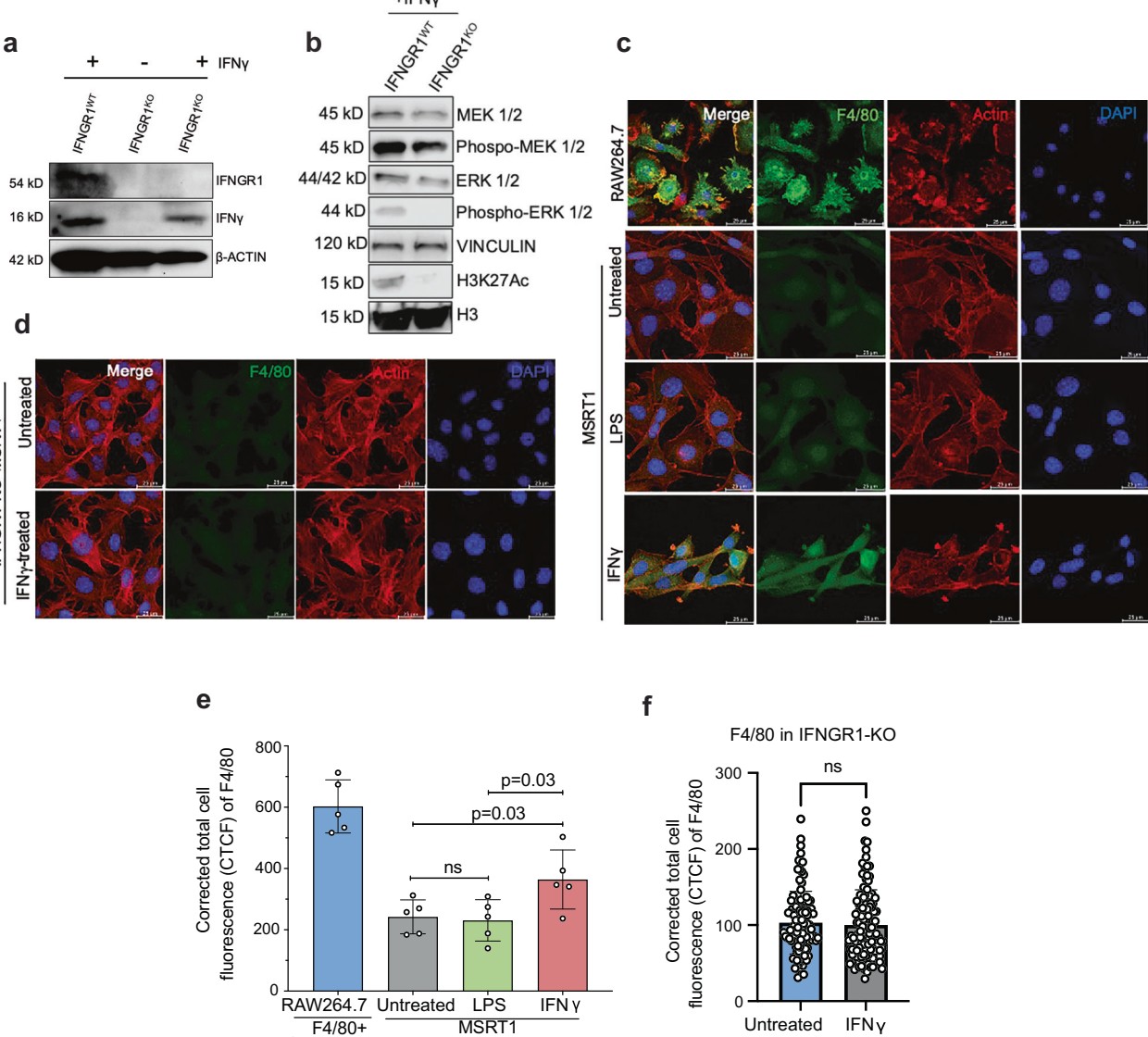

**Fig. 6 | In vitro IFNGR1 knockout in mouse RMC cells. a** Western blot analysis validating the successful knockout (KO) of IFNGR1 in MSRT1, a cell line generated from our RMC mouse model in the C57BL/6 background using CRISPR gene editing technology. **b** Western blot analysis of the MEK/ERK pathway in the MSRT1 cell line demonstrates a significant reduction of the MEK/ERK signaling pathway after genetic knockout of IFNGR1. **c** Representative immunofluorescent images of F4/80, a mouse macrophage marker, in MSRT1 cells treated with interferon gamma (IFNγ) 100 ng/mL or lipopolysaccharides (LPS) 100 ng/mL in vitro. Untreated RAW264.7, a mouse macrophage-like cell line, was used as a positive control for F4/80 expression. Scale bar is 25 μm. **d** Representative immunofluorescent images of F4/80 in MSRT1-IFNGR1[KO] cells treated in vitro with IFNγ 100 ng/mL compared to untreated.

Scale bar is 25 μm. **e** Quantification of F4/80 immunofluorescence in MSRT1 cells treated with IFNγ 100 ng/mL, lipopolysaccharides (LPS) 100 ng/mL, and untreated for 48 h. Untreated RAW264.7, a mouse macrophage-like cell line, was used as a positive control for F4/80 expression. There are $n = 5$ fields per group. **f** Quantification of F4/80 in immunofluorescent study of MSRT1-IFNGR1[KO] cells treated with 100 ng/mL of IFNγ ($n = 118$) compared with untreated ($n = 106$). For (**e**, **f**), each dot represents the number of fields of view of images taken (each field has approximately 50 cells). The experiments were replicated three times, and a representative experiment is shown in (**a**–**d**). Data are expressed as mean value ± SD, with P-value calculated by two-sided student's t test. Source data are provided as a Source Data file.

tumor cells of patients treated with ICT, suggesting that p300 is specifically engaged during RMC hyperprogression.

Our results suggest that p300 engagement and subsequent myeloid mimicry occur via the binding of IFNγ to the IFNGR1 that is overexpressed in RMC tumor cells after ICT, representing a non-canonical downstream pathway of interferon gamma signaling that promotes lineage plasticity. However, targeting of IFNγ / IFNGR1 can directly have pleotropic effects[29], as indicated by our finding that IFNGR1 knockout tumors were significantly larger than IFNGR1 wild-type tumors in mice treated with IgG control. This observation prompted us to investigate whether pharmacologic inhibition of the master myeloid regulator p300 downstream of IFNGR1 could elicit

antitumor immune responses. We developed, characterized, and tested the highly selective orally available CBP/p300 inhibitor IACS-16898 and found that it prevented hyperprogression and induced significant antitumor response to ICT in RMC. These results open therapeutic avenues and prompt the development of future immunotherapy clinical trials combining p300 inhibition with ICT in RMC and potentially other aggressive diseases that may hyperprogress via myeloid mimicry.

Our study is limited by the lack of matched pre- and post-therapy biopsies from the same patient for single-cell analyses. In addition, the multiplex immunofluorescence data findings included only one matched pre- and post-ICT tumor sample from our clinical trial (patient

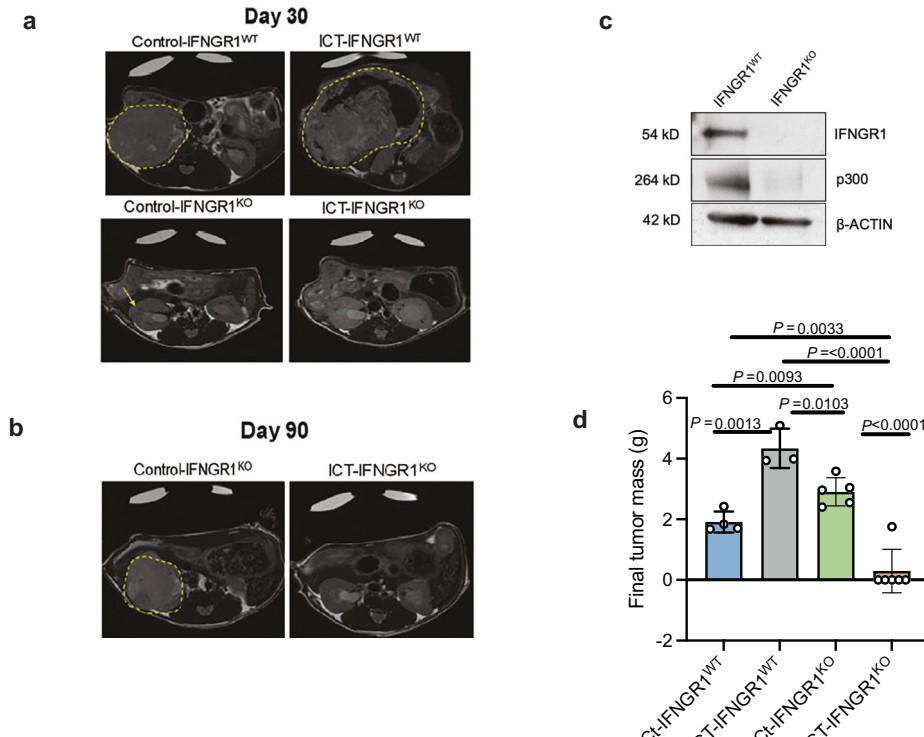

**Fig. 7 | In vivo IFNGR1 knockout in mouse RMC tumors. a** Representative magnetic resonance imaging (MRI) of mice bearing right kidney tumors on day 30 of treatment with either IgG control or anti-PD-1 plus anti-CTLA-4 ICT for mice injected with parental MSRT1 cells and MSRT1 cells with IFNGR1 knockout. Tumor boundaries are traced in the dotted yellow line to illustrate the difference in the size of the control- versus the ICT- treated RMC tumors in the parental (IFNGR1^WT) compared to the genetic knockout (IFNGR1^KO). The yellow arrow indicates tumor that is too small to outline. **b** Representative magnetic resonance imaging (MRI) of mice bearing right kidney tumors on day 90 of treatment with either IgG control or anti-PD-1 plus anti-CTLA-4 ICT for mice injected with MSRT1 cells with IFNGR1 knockout (IFNGR1^KO). IFNGR1^KO cells treated with ICT have almost complete tumor inhibition. **c** Western blot analysis of ex vivo MSRT1 tumors from one representative tumor that was wildtype (WT) for IFNGR1 and one tumor that had IFNGR1 ablation via CRISPR (KO) after 30 days, demonstrating sustained knockout of IFNGR1 in IFNGR1^KO and unexpected loss of p300 protein. **d** Quantification of final masses at time of death for parental MSRT1 (IFNGR1^WT) and IFNGR1 knockout cells (IFNGR1^KO) treated with either IgG control (Ct) or anti-CTLA4 and anti-PD1 (ICT) ($n = 4$ and $n = 3$, $n = 5$, and $n = 6$, respectively). Data are expressed as mean value ± SD, with P-value calculated by two-sided student's t test, and each dot represents one mouse or biological replicate. Source data are provided as a Source Data file.

RMC25). These constraints preclude definitive conclusions about tumor cell–intrinsic transcriptional changes triggered by ICT in patients. Consequently, we relied on functional studies in our genetically engineered mouse model of RMC, which recapitulated the hyperprogression observed in humans and allowed us to interrogate the specific role of IFNγ–IFNGR1–p300 signaling. However, although our genetically engineered mouse model of RMC reproduces the aggressive and metastatic phenotypes observed clinically, there is no perfect equivalence between mouse models and human diseases, and thus, caution must be exercised when extrapolating these findings to the clinic. Although the MSRT1 RMC cell line and its IFNGR1-knockout derivative reproduce key functional axes, they remain murine models and do not entirely mirror the transcriptomic complexity of human RMC. Therefore, conclusions derived from these models should be interpreted with caution and validated in additional human systems. Furthermore, the clinical trial was single-arm and uncontrolled, so we cannot firmly establish causality between the therapy and hyperprogression in patients. While our mechanistic data and mouse model results are compelling, a formal statistical causal inference or randomized controlled design would be required to definitively attribute hyperprogression to the immunotherapy regimen. Additional clinical studies with fully matched pre- and post-therapy biopsies are needed to validate these findings in patients.

Notably, a meta-analysis of synovial sarcoma, another neoplasm driven by deregulation of *SMARCB1*, revealed that high expression of p300 expression was associated with poor prognosis[44]. Other studies have also shown that high expressions of p300 is associated with poor prognosis in several other solid tumors, including non-small cell lung cancers, prostate cancers, breast cancers and laryngeal squamous cell carcinoma subsets[45–49]. In the present study, we leveraged the well-established molecular and clinical homogeneity of RMC[1,9], which allowed us to clearly detect an unusually prominent clinical signal of hyperprogression in our prospective clinical trial and subsequently replicate and functionally characterize these findings in our RMC mouse model. Our findings open avenues to identify cancer subtypes that hyperprogress in response to ICT by hijacking myeloid-affiliated pathways and thus allows for the development of effective combination therapies by targeting this mechanism of hyperprogression, and also support the emerging strategy of combining epigenetic modulators with ICT to prevent tumor-promoting epigenetic processes in cancers that are resistant to ICT[50] and possibly other therapies.

## Methods
### Trial oversight
The trial was designed and conducted at the University of Texas MD Anderson Cancer Center (MDACC), registered under clinicaltrials.gov identifier NCT03274258, approved by the MDACC institutional review board (IRB)/independent ethics committee (IRB protocol 2017-0201) and conducted in accordance with Good Clinical Practice guidelines, defined by the International Conference on Harmonization. All patients provided written informed consent to participate based on the principles of the Declaration of Helsinki. The 10 patients enrolled in the prospective clinical trial (Table 1) provided informed consent under the trial IRB protocol 2017-0201 and IRB protocol PA11-1045. The

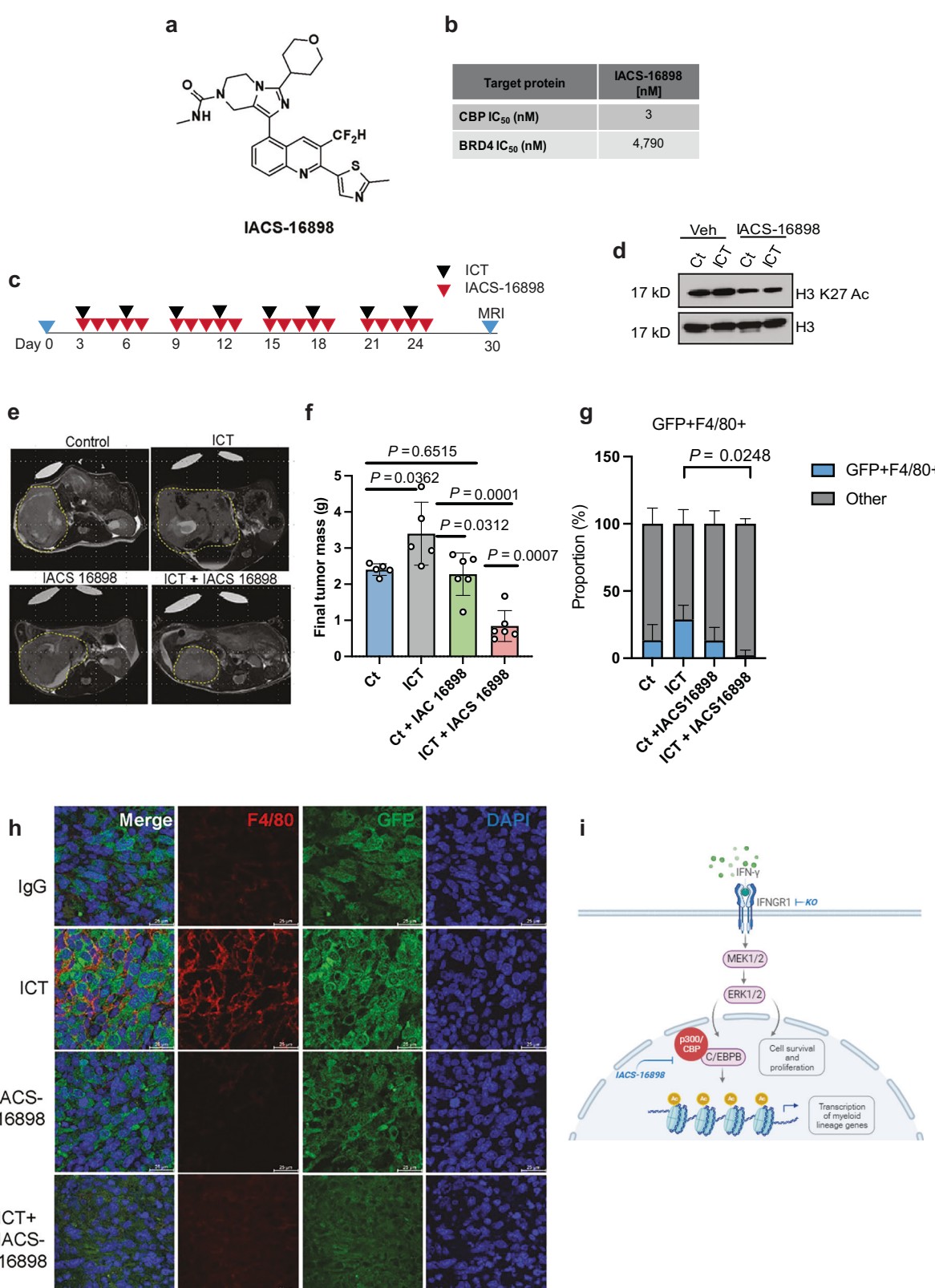

seven additional patients who participated in the single-cell RNA sequencing cohort (Supplementary Table S7) provided informed consent under IRB protocols PA11-1045 and PA17-0577. This manuscript adheres to the ICMJE recommendations for the conduct, reporting, editing, and publication of scholarly work in medical journals. The full trial protocol is available as Supplementary Note in the Supplementary Information.

## Patients

The first and last patient started trial therapy on June 7th 2018, and December 2nd 2020, respectively. Participants received no financial compensation for taking part in this study. Findings apply to both sexes. Sex was not considered in the study design. Both males and females were eligible for the study. Sex was determined based on self-reporting. Eligible patients were ≥ 18 years old with locally advanced or

**Fig. 8 | Pharmacological perturbation of master regulatory pathways of myeloid mimicry. a** IACS-16898 structure. **b** AlphaScreen results for specific binding of the CBP or BRD4 bromodomain. **c** Schema showing treatment schedule with ICT and IACS 16898 over the course of 30 days. **d** Western blot analysis of global acetyl-H3K27 (H3K27Ac) on total histones extracted from one representative tumor from each treatment cohort at day 30. **e** Representative magnetic resonance imaging (MRI) of mice bearing right kidney tumors on day 30 of treatment with either IgG control or anti-PD-1 plus anti-CTLA-4 ICT in combination with IACS 16898, an inhibitor of CBP/p300 for mice injected with parental MSRT1 cells. Tumor boundaries are traced in the dotted yellow line to illustrate the difference in the size of the control- versus the ICT (anti-PD1 and anti-CTLA4)- treated RMC tumors in combination with 5 mg/kg of IACS 16898, a CBP/p300 inhibitor. **f** Quantification of tumor mass collected at day 30, demonstrating that inhibiting CBP/p300, a master regulator of myeloid differentiation, restores sensitivity to ICT. Ct, $n = 5$; ICT, $n = 5$;

Ct + IACS 16898, $n = 6$; ICT + IACS 16898, $n = 6$. Data are expressed as mean value ± SD, with $P$-value calculated by two-sided student's $t$ test, and each dot represents one mouse or biological replicate. **g** Parts of the whole of the percentage of tumor cells co-expressing GFP and F4/80 in RMC mouse models treated with IgG control (Ct, $n = 3$), anti-PD-1 plus anti-CTLA-4 (ICT, $n = 3$), IgG plus IACS-16898 ($n = 3$), and ICT plus IACS-16898 ($n = 3$). Three biological replicates were selected for analysis per group. **h** Representative immunofluorescent images of F4/80, a mouse macrophage marker, co-expressed with GFP (tumor marker) in one representative ex vivo MSRT1 tumor from each treatment cohort after 30 days of treatment with IgG, ICT, IACS-16898, and ICT + IACS-16898. Scale bar is 25 μm. Images are 40X magnification. **i** Schema of signaling pathway causing myeloid mimicry and hyperprogression in RMC tumors in response to ICT. Created in BioRender. Yan, X. (2025) https://BioRender.com/vqkcx1g. Source data are provided as a Source Data file.

metastatic RMC confirmed to be negative for SMARCB1 by Clinical Laboratory Improvement Amendments (CLIA) certified immunohistochemistry (IHC) using purified mouse anti-BAF47 Clone 25/BAF47 (BD Biosciences). Sickle cell status is determined by hemoglobin electrophoresis for all patients enrolled with a diagnosis of RMC. Patients with RMC without sickle hemoglobinopathies were also eligible. Cytoreductive nephrectomy is the recommended approach for patients responding to systemic therapy and is thus allowed in our clinical trial following adequate response to the trial regimen (Supplementary Fig. S1)[1]. Patients had an Eastern Cooperative Oncology Group (ECOG) performance status score of 0, 1, or 2, as well as measurable disease by the Response Evaluation Criteria in Solid Tumors (RECIST version 1.1)[51], and could either naïve for any previous systemic treatment or have had any number of prior systemic therapies. However, patients must not have received prior anticancer therapy with anti-PD1, anti-PD-L1, or anti-CTLA-4 immune checkpoint inhibitors. Additional key exclusion criteria included known or suspected history of autoimmune disease requiring systemic immunosuppression, current use of an immunosuppressant (> 10 mg daily prednisone equivalent) or other more potent immune suppression medications such as infliximab, uncontrolled brain or leptomeningeal metastases, known history of human immunodeficiency virus (HIV) or known acquired immunodeficiency syndrome (AIDS), and a positive test for hepatitis B virus (HBV) using HBV surface antigen (HBVsAg) test or positive test for hepatitis C virus (HCV) using HCV ribonucleic acid (RNA) or HCV antibody test indicating acute or chronic infection. Refer to the full trial protocol in the supplementary note for more details.

## Clinical trial design

This was an investigator-initiated, phase 2, open-label, single-arm trial testing the efficacy of combination nivolumab plus ipilimumab in patients with locally advanced or metastatic RMC. The combination dosing scheme chosen followed that of the phase 3 CheckMate 214 clinical of nivolumab plus ipilimumab in clear cell renal cell carcinoma[52]. Nivolumab was accordingly administered at a dose of 3 mg/kg every 3 weeks in combination with ipilimumab 1 mg/kg every 3 weeks for up to 4 doses, followed by maintenance with single-agent nivolumab 480 mg flat dose every 4 weeks until disease progression or intolerable toxicity. The maintenance nivolumab flat dose of 480 mg every 4 weeks has been shown to produce similar exposure/response outcomes compared with every 2-week dosing schedules across multiple tumor types based on quantitative clinical pharmacology analyses and safety assessments[53].

## Clinical trial endpoints and assessments

The primary objective was to estimate the objective response rate (ORR) of patients with locally advanced or metastatic RMC treated with a combination of nivolumab plus ipilimumab. ORR is defined as the proportion of patients with a best response of complete response (CR) or partial response (PR) by the RECIST 1.1 criteria recorded

between Day 1 of the study and the date of objectively documented progression per RECIST 1.1 or the date of subsequent anti-cancer therapy, whichever occurred first. The goal was to significantly improve the ORR compared with the historical ORR of 29% achieved using conventional cytotoxic chemotherapies[9]. Secondary objectives included determining the efficacy and safety of the combination of nivolumab plus ipilimumab in patients with RMC, with efficacy measured by overall survival (OS), progression-free survival (PFS), time to ORR, duration of response, and the disease control rate (DCR).

The trial was originally planned to enroll up to 30 patients based on the feasibility of completing the trial within 3 years while retaining acceptable operating characteristics for the monitoring described (Supplementary Tables S12, S13). One hypothesis test was planned to compare the historical response rate of 29% (13/45) to the posterior distribution of this trial's response rate. This would be tested using the posterior probability that the beta-binomial posterior distribution from this trial was greater than the beta-binomial distribution from the historical trial. This hypothesis was only planned to be tested if the trial successfully accrued to 30 patients. Denote the probability of objective response rate (ORR) in this trial by $\theta_R$ and the probability of ORR in the historical cohort by $\theta_H$. If this trial had 12 or more responses, then $\Pr(\theta_R > \theta_H \mid \text{data})$ would be greater than 0.80, using posterior probabilities of $\theta_H \sim \text{Beta}(13,32)$ and $\theta_R \sim \text{Beta}(r + 0.6, q + 1.4)$ where r is the number of responders and q is the number of non-responders ($q = 30$-r) and the prior for $\theta_R$ is beta(0.6, 1.4).

Ongoing monitoring for safety and futility were implemented based on the method of Thall et al.[54] using cohorts of 10 patients each. Accrual was held between cohorts to assess for both toxicity and response. Calculations were performed in Multc Lean 2.1 [available at https://biostatistics.mdanderson.org/SoftwareDownload/]. Extreme toxicities (TOX) were defined as any event requiring the patient to be removed from the trial according to predefined rules. A patient who voluntarily left the trial for any reason, including toxicity that did not meet the protocol criteria, did not count as having a TOX. A TOX could occur any time during the study. For cohort monitoring, the most recent patients were classified by their status at the time of cohort analysis, so the shortest follow-up time was expected to be 12 weeks. Denote the probability of TOX by $\theta_T$. Our stopping rule was given by the following probability statement: $\Pr(\theta_T > 0.30 \mid \text{data}) > 0.95$. That is, the trial would stop if, at any time during the study, we estimated that there was more than a 95% chance that the TOX rate is more than 30%. These rules assumed a TOX rate for "standard therapy" of 0.30 and a prior $\theta_T \sim \text{beta}(0.6, 1.4)$ for the current study. The stopping boundaries for this toxicity rule were to terminate the trial if the number of patients with TOX compared to the number of patients having received treatment exceeds the limits in Supplementary Table S12.

Similarly, for futility monitoring, the ORR was determined by CR or PR by RECIST at 12 weeks. Denote the probability of ORR by $\theta_R$. We assume $\theta_R \sim \text{beta}(0.5, 0.5)$. It is standard practice in Bayesian trial design methodology[55] to use a community of priors, also known as

"adversarial" priors, whereby an enthusiastic prior is used for futility monitoring while a more skeptical prior is used for the final analysis. The goal is to counterbalance the prior opinions of someone who would doubt the observed results. Thus, if even an enthusiastic prior indicates futility at interim analysis, then the trial should be stopped. The evidence of futility should be sufficient to convince a reasonable enthusiast. Conversely, the final analysis should be able to convince more skeptical readers. In the present trial, the interim futility monitoring prior beta (0.5,0.5) conveys high uncertainty without conveying skepticism. It is thus more enthusiastic than the final analysis prior of beta (0.6, 1.4), which represents a more skeptical view about the success probability, leaning toward values close to 0. Our stopping rule was given by the following probability statement: $\Pr(\theta_R < 0.30 \mid \text{data}) > 0.90$. That is, we would stop entering patients if, at any time during the study, we estimated that there is more than a 90% chance that the ORR was less than 30%, a constant rate that is slightly above the expected response rate for these patients on other treatments. The stopping boundaries for this futility rule were to terminate enrollment on the trial if the number of overall responses compared to the number of patients enrolled who were evaluated did not meet the requirements of Supplementary Table S12. Patients who left the study after receiving treatment but before the 12-week evaluation were counted as non-responders and therefore evaluable for response. Patients who enrolled but never received treatment were planned to be replaced. Following early termination of the trial for futility, we additionally performed a sensitivity analysis using the more skeptical beta (0.6, 1.4) prior, which resulted in identical stopping rules as those shown in Supplementary Table S12 and would have therefore not changed the trial outcome.

**Definition of hyperprogression.** Hyperprogression was defined using the previously established criteria for solid tumors as progressive disease in the first 8 weeks after treatment initiation, a 10 mm minimum increase in the measurable lesions, plus: (i) an increase of ≥ 40% in the sum of target lesions compared with baseline and/or (ii) an increase of ≥ 20% in the sum of the target lesions compared with baseline plus the appearance of new lesions in at least two different organs.

## Clinical trial analysis

As prespecified in the trial protocol (see full protocol in the Supplementary Information), descriptive statistics are presented. For discrete or categorical data, including ORR and DCR, descriptive statistics include tabulations of frequencies. For continuous data, summary statistics including n, median, minimum, and maximum were computed. The PFS and OS are presented with Kaplan-Meier curves and 95% confidence bands. Patient adverse events and the number of patients required to be removed from the study are tabulated by symptom grade and attribution to study drug. All analyses were performed in SAS 9.4 [SAS Institute Inc., Cary, NC], and figures were generated in Stata 17 [StataCorp LLC, College Station, TX].

## Bulk RNA sample preparation and analysis

**RNA extraction and quality control.** Total RNA was isolated using the NORGEN Total RNA Purification Kit (Cat. 37500) (NORGEN BIOTEK CORP). Residual genomic DNA was removed by DNase I treatment, and RNA samples were subsequently purified with AMPure XP beads (Beckman Colter Life Sciences) and eluted in 1 × TE buffer. RNA concentration was determined with the Quant-iT™ RiboGreen™ RNA Assay Kit (ThermoFisher SCIENTIFIC), and integrity was verified using the Agilent RNA 6000 Nano Kit on a 2100 Bioanalyzer Instrument (Agilent Technologies).

**cDNA Synthesis.** Complementary DNA (cDNA) was generated from total RNA using the Ovation® RNA-Seq System V2 (NuGEN), which initiates amplification both from polyadenylated regions and random priming sites to capture broad transcriptome coverage. cDNA quantity and quality were evaluated using the Quant-iT™ PicoGreen™ dsDNA Assay Kit (ThermoFisher Scientific) and the TapeStation 4200 (Agilent Technologies) with Genomic DNA ScreenTape reagents.

**Library preparation.** For each sample, up to 200 ng of cDNA (as determined by PicoGreen quantification) was fragmented using a Covaris E220 Focused Ultrasonicator (Covaris) with the following parameters: 200 peak incident power, 25% duty cycle, 50 cycles per burst, and 10 s duration repeated 120 times.

Up to 200 ng of each cDNA sample, based on the PicoGreen quantification, was sheared (mechanically fragmented) using the E220 Focused-ultrasonicator Covaris (Covaris). The sonication was performed under the following conditions: Peak Incident Power 200, Duty Cycle 25%, Cycles per Burst 50, and duration 10 s for 120 iterations. Fragment size was confirmed on the TapeStation 4200 using the High Sensitivity DNA kit (Agilent). Library construction was carried out using the SureSelect XT Low Input Reagent Kit with barcoded indexes (Agilent Technologies) on a Sciclone G3 NGSx Workstation (PerkinElmer, Inc.).

End repair, A-tailing, and adapter ligation were performed enzymatically, followed by PCR amplification with Herculase II Fusion DNA Polymerase (8–14 cycles depending on input quality). Amplicons were purified using Agencourt AMPure beads (Agencourt Bioscience Corporation), and library size distribution was verified on the TapeStation 4200 and the DNA High Sensitivity kit (Agilent Technologies) to confirm correct fragment size and removal of primer dimers.

**Hybridization and capture.** Indexed libraries were hybridized with Agilent SureSelect Human All Exon v4 probes (Agilent Technologies) following the manufacturer's instructions, automated on the Sciclone G3 NGSx Workstation (PerkinElmer, Inc.). Captured fragments were retrieved with streptavidin-coated beads, washed, and enriched by post-capture PCR. Enriched libraries were again assessed for size and concentration using the TapeStation 4200.

**Sequencing and data analysis.** Final libraries were sequenced on the Illumina NovaSeq 6000 platform (Illumina) to produce paired-end 150 bp reads using v3 reagents.

**Raw data.** Raw RNA sequencing data were demultiplexed and converted to FASTQ format using Illumina's Consensus Assessment of sequence and Variation (CASAVA) (http://support.illumina.com/sequencing/sequencing_software/casava.html). FASTQ files were aligned to the reference genome with STAR (two-pass mode)[56], and TopHat[57], and transcript quantification was performed using Cufflinks[58].

**Data analysis.** Aligned BAM files were analyzed for germline and somatic variants using Platypus[59], MuTect[60], and Pindel[61]. Structural rearrangements and fusions were identified with Delly[62], TopHat[57], MapSplice[63], SOAPFuse[64] or Spladder[65]. Additional RNA quantification is conducted using HTSeq[66] analyses.

**Differential expression and functional enrichment analysis.** Differential gene expression was analyzed using DESeq2 (v.1.34.0)[67], considering genes with |log₂FoldChange| > 1 and adjusted $p < 0.05$ as significant. Gene Set Enrichment Analysis (GSEA) was performed with clusterProfiler (v.4.2.2)[68] using Reactome pathways from MSigDB (v.2023.1.Hs)[69]. Enriched pathways (|NES| ≥ 1, $p < 0.05$) were visualized, highlighting biological processes related to cell cycle regulation and immune response.

## ScRNA-seq data processing and analysis

**Tissue samples.** We obtained samples at baseline prior to nivolumab plus ipilimumab initiation from 7 patients with RMC and following

progression on nivolumab plus ipilimumab from 2 patients with RMC. Supplementary Table S3 lists the timepoint and specific organ site biopsied for each patient sample.

**scRNA-seq library preparation and sequencing.** Fresh tumor specimens were finely minced (<1 mm) and enzymatically digested in RPMI-1640 containing 0.1 mg/mL Liberase TM and 0.2 mg/mL DNase I (MilliporeSigma) for 15 min at 37 °C with gentle agitation. The resulting suspensions were filtered through 70 μm strainers, pelleted, and subjected to red blood cell lysis with ACK lysis buffer (ThermoFisher, catalog #A1049201). Cell viability was determined by trypan blue exclusion, and only samples exceeding 60% viable cells were processed for single-cell encapsulation on the Chromium iX instrument (10x Genomics) to generate 5' v2 gel beads-in-emulsions (GEMs). Barcoded cDNA was amplified, purified, and quantified on an Agilent 2100 Bioanalyzer before sequencing at ~ 50,000 reads per cell on an Illumina NovaSeq 6000 platform (paired-end 150 bp).

**Single-cell RNA data preprocessing and filtering.** Sequencing data were demultiplexed, aligned to the GRCh38 reference genome, and quantified using CellRanger v6.0.1.(10X Genomics). Downstream analyses were performed in Seurat v4.3.0[70]. Cells expressing fewer than 200 genes or with over 20% mitochondrial reads were excluded to remove low-quality or dying cells; genes detected in fewer than three cells were also discarded. Cells with extremely high gene counts (> 6500) were removed to minimize the inclusion of multiplets. After quality filtering, 23,880 single cells were retained for downstream analysis.

**Clustering and major cell type annotation in scRNA-seq.** Normalized and scaled expression matrices were used for dimensionality reduction with principal component analysis (PCA). The first 30 principal components were selected for unsupervised clustering using Seurat's *FindNeighbors* and *FindClusters* functions at a resolution of 0.3, producing 22 initial clusters. The reduced dimensions were visualized with UMAP[71] embeddings. Potential doublets were manually identified and removed based on the co-expression of markers from distinct lineages or atypically high gene counts.

The filtered gene-cell matrix were processed using the Seurat package (v.4.3.0)[70], which included normalization using the *NormalizeData* function to obtain normalized UMI (unique molecular identifier) count data, scaling, and centering the data using *ScaleData*, finding the most significant principal components (PCA) using *RunPCA* and using the *ElbowPlot* to determine the number of principal components used for clustering. Different resolution parameters for unsupervised clustering were then examined to determine the optimal number of clusters. The first 30 principal components were used for unsupervised cell clustering with a resolution = 0.3, yielding a total of 22 cell clusters using the *FindNeighbours* and *FindClusters* functions. For visualization, the dimensionality was further reduced using the UMAP methods with the Seurat function *RunUMAP*. The same principal components used for clustering were used to calculate the UMAP embedding.

Potential doublets were further identified and removed from each resulting cluster. Doublets were identified by the following standards: (1) Cells within a cluster express canonical markers from different lineages (for example, cells in the T cell cluster showed expression of epithelial cell markers); (2) doublets or multiplets likely form distinct clusters with hybrid expression features and exhibit an aberrantly high gene count. We carefully reviewed canonical marker gene expression on UMAP plots and repeated the steps above a couple of times to ensure that we had filtered out most of the barcodes associated with cell doublets. Based on the above criteria and removing doublets, a total of 23,880 cells were kept for downstream analyses. Cluster annotation was performed by examining the top differentially expressed genes (FindAllMarkers function) and the enrichment of canonical lineage markers, including *EPCAM* (epithelial), *PTPRC* (immune), *CD3D/E* (T cells), *CD19/MS4A1/CD79A* (B cells), and *COL1A1/COL1A2* (fibroblasts). We accordingly generated dot plots for a set of canonical immune and stromal cell markers (Supplementary Fig. S2d). Ten major cell types were defined, and T/NK subsets were further resolved into CD8+, CD4+, and NK transcriptional states.

**Identification of malignant cells.** To distinguish malignant from non-malignant epithelial cells, large-scale chromosomal copy-number variations (CNVs) were inferred using InferCNV (https://github.com/broadinstitute/inferCNV) as previously described[15,72–74]. T cells served as the reference population. CNV scores were computed as the average squared deviation of expression across genomic bins[75], and k-means clustering grouped epithelial cells according to their CNV patterns (Supplementary Fig. S4). Clusters exhibiting high CNV scores were classified as malignant.

**Differential gene expression and Pathway Enrichment Analysis.** Within each major cell type (malignant epithelial, CD8+ T, CD4+ T, NK, and myeloid), differential expression between post-treatment and baseline samples was assessed using the *FindMarkers* function in Seurat (v.4.3.0)[70]. Pathway-level analysis was performed with GSEA based on Reactome pathways. Enriched pathways associated with proliferation and immune signaling were highlighted for visualization.

**Similarity and distance metrics.** To assess transcriptional proximity between tumor and myeloid populations, six metrics were computed: Euclidean, Manhattan, Bray–Curtis, and Phi dissimilarities, as well as cosine and Pearson similarities. Calculations were based on the top 3000 highly variable genes using functions from the R packages stats (v.4.1.0). Cosine similarities were calculated in the 'lsa' package (v.0.73.1)[76]. Bray-Curtis was calculated using the 'vegan' package (v.2.7-1)[77], and Phi was calculated using the 'propr' package (v.5.1.7)[78]. Group differences were tested with the two-sided Wilcoxon rank-sum test, and combined violin/box plots were generated using the geom_split_violin from the R package introdataviz (v.0.0.0.9003) and geom_boxplot function from the R package ggplot2 (v.3.5.1).

**Cell-cell communication analysis.** Intercellular signaling networks were inferred using the 'CellChat' package (v.2.1.2)[24].

**Tumor cell entropy analysis.** Tumor cell entropy was estimated using the single-cell signaling entropy method SCENT[21].

## CyTOF staining
Metal-conjugated antibodies were purchased from Standard BioTools for available targets. For all other targets, in-house antibody conjugation of commercially available primary antibodies were prepared using the Maxpar X8 Antibody Labeling Kit (Standard BioTools, South San Francisco, CA, USA) per manufacturer instructions (Supplementary Table S14). All antibodies were titrated before use.

Prior to staining, cryopreserved PBMCs were thawed in a water bath at 37 °C and washed twice in thawing media (RPMI 1640 and 10% inactivated FBS) containing 1:10,000 Pierce Universal Nuclease (Thermo Scientific, Waltham, MA, USA). The cells were assessed for post-thaw counts and viability using an automated cell counter (Nexcelom Bioscience, Lawrence, MA, USA). $2 \times 10^6$ cells from each sample were aliquoted for CyTOF staining. For viability staining, cells were stained with Cell-ID cisplatin 198Pt (Standard BioTools) at a final concentration of 5 μM for 5 min at RT. Prior to antibody staining, samples were incubated with Human TruStain FcX (Biolegend, San Diego, CA, USA) for 10 min at RT for Fc-receptor blocking. To eliminate sample-specific staining variation, all samples were barcoded and then stained antibodies, processed and acquired as one multiplexed sample. Samples were then barcoded using a 20-plex CD45 Live-cell barcoding kit

(Standard BioTools) for 30 min at RT. Each sample was washed twice with Maxpar Cell Staining Buffer after incubation with different barcodes, and all samples were combined into one tube. Next, cells were stained for 30 min at RT with the cocktail of surface anitbodies. After the surface stain incubation, samples were washed twice using Maxpar Cell Staining Buffer, then fixed and permeabilized with Foxp3/Transcription Factor Staining Buffer Set (eBioscience) and intracellular staining was performed for 45 min at RT. The staining was followed by washing twice in 1X Perm Wash and fixed with 1.6% paraformaldehyde (Thermo Fisher) for 10 min at RT. Afterwards, the intercalation solution was prepared by adding Cell-ID Intercalator-Ir into Maxpar Fix & Perm Buffer (Standard BioTools) to a final concentration of 41.6 nM (a 3000× dilution of the 125 μM stock solution). After the second fixation, the cells were resuspended with the intercalation solution and incubated overnight at 4 °C. Immediately prior to data acquisition, samples were then washed with Maxpar Cell Staining Buffer and then with subsequent washes in Maxpar Water (Standard BioTools) to remove buffer salts. Next, the cells were resuspended at a concentration of $1 \times 10^6$ cells/mL in Maxpar Water containing a 1:10 dilution of EQ Four Element Calibration Beads (Standard BioTools) and filtered through a 35 μm nylon mesh filter cap (Corning, Falcon). The samples were then acquired on a Helios Mass Cytometer equipped with an HT sample injector at an acquisition rate of < 500 events/s.

### CyTOF data analysis

Mass cytometry data were normalized, concatenated, and de-barcoded using CyTOF Software v.7.0 (Standard BioTools, South San Francisco, CA, USA). FCS files were manually processed in FlowJo v10 (TreeStar, Ashland, OR, USA) to exclude Ce140+ beads, Gaussian ion cloud fusion events, debris, Pt198 + dead cells, doublets (Supplementary Fig. S9) and exported into new FCS files, which was used for downstream analysis[79].

We used a combination of traditional manual gating validation (Supplementary Fig. S10) and unbiased approaches to analyze our datasets. A total of 10,000 live singlets from each sample were randomly selected for clustering and dimensionality reduction algorithms. All marker expressions (raw data) were arcsinh-transformed with a cofactor of 5 (counts_transf = asinh(x/5)). Cell population identification was carried out by using the FlowSOM[80] packages in R version 4.2.2, using 28 lineage markers and 20 metaclusters. Clusters were manually identified based on the heatmaps with normalized to 0-1 median marker expression. For data visualization, the tSNE dimension reduction was performed using Rtnse package, to represent the annotated cell populations in a 2D map[81]. The proportion of each cell population abundance was exported to GraphPad Prism Version 9 (GraphPad Software, San Diego, CA, USA) to create graphs and statistics. Data presented as mean ± SEM. *P*-values were calculated by a non-parametric Mann-Whitney U test, and *p*-values less than 0.05 were considered significant. The major cell types identified from the CyTOF (mass cytometry) analysis are shown in Supplementary Figs. S11–S13.

### Immunoblotting/Western blotting

Cells were lysed with RIPA buffer containing protease inhibitor. Protein lysates were resolved on 5–15% gradient polyacrylamide SDS gels and transferred onto PVDF membranes according to the manufacturer's instructions. Membranes were incubated with the indicated primary antibodies, washed in TBST buffer and probed with HRP-conjugated secondary antibodies. The detection of bands was carried out upon chemiluminescence reaction, followed by film exposure. The antibodies used are listed in Supplementary Table S15. Uncropped scanned films are provided in Supplementary Figs. S33–S37.

### Multiplex Immunofluorescence for patients staining and analysis (automated using Leica Bond)

Multiplex immunofluorescence (mIF) staining was performed using a similar method previously described and optimized[82–84]. Briefly,

consecutives serial sections of four micrometer-thick formalin-fixed, paraffin-embedded sample were stained using an automated staining system (BOND-RX; Leica Microsystems, Buffalo Grove, IL) using four mIF panels: Panel 1, pancytokeratin (panCK), CD3, CD8, PD-1, PD-L1, FOXP3, CD68, and KI67; Panel 2, panCK, CD3, CD8, PD-1, PD-L1, FOXP3, CD20, and CD68; Panel 3, panCK, CD3, ICOS, LAG3, TIM3, VISTA, and OX40; and Panel 4, panCK, CD68, MRP8-14, CD86, CD206, CD163, Arg-1, and PD-L1 (Supplementary Table S16). All the markers form each panel were stained in sequence using their respective fluorophore containing in the Opal 7 IHCkit (catalog #NEL797001KT; Akoya Biosciences, Marlborough, MA) and the individual tyramide signal amplification fluorophores Opal Polaris 480 (catalog #FP1500001KT) and Opal Polaris 780 kit (catalog #FP1501001KT, Akoya Biosciences). The slides were scanned using the Vectra Polaris 1.0.13 (Akoya Biosciences) at low magnification, 10x (1.0 μm/pixel) through the complete emission spectrum and using positive tonsil controls from the run staining to calibrate the spectral image scanner protocol[85]. The analysis was made by a pathologist in five intratumoral areas using $930 \times 698$ (0.65 mm²) region of interest (ROI) at x 20 magnification to cover a total intratumoral area of 3.25 mm². When the five ROIs don't cover 3.25 mm² of intratumoral area, the pathologist performed the analysis in more than five regions of interest (ROIs), to cover at least 3.25 mm² if there are more intratumoral areas available; if not, we performed the analysis in the entire intratumoral areas available using the InForm 2.4.8 image analysis software (Akoya Biosciences). Using the panCK) marker, the intratumoral area from each panel was compartmentalized into epithelial (tumor nets) and stroma compartments (tissue between tumor nets). The data is divided into tumor compartment, stroma compartment and total (tumor and stromal together). Marker co-localization was used to identify specific tumor and stroma compartment cell phenotypes. The densities of each cell phenotype were quantified, and the final data was expressed as the number of cells/mm2. The data were consolidated using R 3.5.3 (Phenopter 0.2.2 packet; https://rdrr.io/github/akoyabio/phenoptrReports/f/, Akoya Biosciences).

### Multiplex Immunofluorescence for mouse and patient tissue staining and analysis (manual staining)

Mouse and patient FFPE (formalin fixed paraffin embedded) tissue sections were used for multispectral staining with the Akoya Biosciences® Opal 3-plex manual detection kit (NEL810001KT). Additional opal dyes were purchased separately to achieve higher plex staining panels. Slides with 5-micron tissue sections were baked in an oven at 60 degrees Celsius for 1 h. Slides were then dewaxed using xylene (3 × 10 min), then rehydrated using an ethanol gradient of 100% (5 min), 95% (5 min) then 70% (2 min) ethanol, anhydrous. Slides were then washed with MilliQ-water then fixed in 10% neutral buffered formalin for 20 min. To perform antigen unmasking, slides were then placed in a plastic Coplin jar and filled to the top with AR6 (AR600250ML) or AR9 (AR900250ML) buffers from Akoya Biosciences®. The entire jar and slides underwent microwave treatment of 55 s at 100% power, then 15 min at 20% power using a Panasonic Inverter microwave for the purpose of antigen retrieval. Slides were left at room temperature afterwards to cool for approximately 20 min and then washed briefly in MilliQ-water followed by 1 x Tris Buffered Saline with Tween-20, 0.05% (1x TBST). Using a PAP pen, a hydrophobic barrier was drawn around the tissue sections to prepare the sections for incubation. A humidified chamber was prepared for all the subsequent incubation steps. Blocking was then performed by incubating each section with 100–200 μL of the antibody blocking reagent provided by the Opal kit (NEL810001KT) for 30 min at room temperature. To remove the blocking solution, each slide was shaken onto a paper towel or absorbent pad; the primary antibody of interest, prepared in SignalStain antibody diluent (Cell Signaling, 8112S), was then applied at 200 μL to each tissue section for overnight incubation

at 4 degrees Celsius or 2 h at room temperature. Three mIF panels were stained with primary antibodies: Panel 1 (patient samples), IFNGR1, S100A9, Keratin 19, IFNG; panel 2 (preclinical mouse samples), GFP, S100A9; panel 3 (preclinical mouse samples), GFP, CD68 (Supplementary Table S17). To block endogenous peroxidases and reduce background staining, slides were then incubated in 1% hydrogen peroxide for 10 min. Subsequent steps are for the purpose of signal generation. Slides were incubated in secondary Polymer HRP Mouse or HRP Rabbit (Opal kit) for 10 min at room temperature, then washed 2 times for 2 min each in 1X TBST. Opal dyes were prepared at 1:200 in 1X plus amplification diluent (Opal kit). Opal™ dyes used were Opals 520, 570, 620, 650, and 690. Slides were incubated for 10 min in Opal Dye at room temperature, then washed 3 times for 2 min each in 1X TBST. Microwave treatment was repeated to remove primary and secondary antibodies from the previous incubation and expose the antigens for the next round of incubation. All steps following microwave treatment were then repeated until the slides had been incubated with all antibodies of interest. After the signal generation for the last antibody was completed, the slides underwent microwave treatment once more to remove any nonspecific binding. Slides were cooled at room temperature, and rinsed as before, then incubated with 1-3 drops (30–100 μL) each of Spectral DAPI (Opal kit) for 5 min. The DAPI solution was then removed, slides were rinsed with 1X TBST (3 times for 2 min each), then coverslipped with Vectamount® aqueous mounting medium (H-5501). Finally, to ensure the signal and quality of the slides are maintained over a long period of time, the slides were sealed with clear nail polish. Images were taken with the Vectra® 3 or Polaris automated quantitative pathology imaging system. Five to ten representative images were snapped for each tissue section with the 20x objective lens. The Inform® 2.6 software from Akoya Biosciences® was used for unmixing and quantification of the captured images.

### Immunofluorescence in vitro experiment

Mouse kidney cancer cell line (MSRT1) and mouse macrophage-like cell line (RAW264.7) were cultured in DMEM media and RPMI media supplemented with 10% FBS. The IFNGR1 KO MSRT1 cell line was cultured in MSRT1 cell line media, but in the presence of Puromycin. IFNGR1 KO in the MSRT1 cell line was validated using western blot. For Immunostaining, these cells were either left untreated or treated with lipopolysaccharides (LPS) 100 ng/mL or interferon gamma (IFN-γ) 100 ng/mL for 48 hrs. Then the cells were fixed in paraformaldehyde for 15 mins and permeabilized in Tritonx100 for 20 min. Cells were then washed with 1X PBS and then blocked in Bovine Serum Albumin (BSA) for an hour followed by overnight incubation with rabbit F4/80 antibody (cell signaling technology, clone D2S9R, cat # 70076), rat Ly6c antibody (BioRad, clone ER-MP20, cat # MCA2389GA), rabbit CD11b antibody (Abcam, clone EPR1344, cat # ab133357), and rabbit CD11c antibody (cell signaling technology, clone D1V9Y, cat # 97585) at 4 °C. The next day, cells were washed three times with 1X PBS and then incubated with Alexa fluor anti-rabbit or anti-rat secondary antibody for an hour at RT, followed by three times washing with 1X PBS and then counter stained with DAPI and Phalloidin to visualize DNA and actin filaments, respectively. Cells were then mounted on a glass cover slide and then observed under the microscope for imaging.

For acquisition of the high-resolution images, an HC PL APO 100 ×/1.40 CS2 laser scanning confocal microscope (Leica Sp8) with spectral emission detection on an inverted stand, fully motorized stage, fast z movement (Leica Super Z Galvo stage) with resolution up to ~ 20 nm was used that allowed observation of greater details of subcellular structures. For quantification of F4/80 protein expression, exposure time was kept unchanged across all conditions. All the acquired images were then processed using Leica's LASX software.

The acquired fluorescent images were then analyzed using FIJI software (Version 2.0.0-rc-49/1.51a) by selecting one cell at a time in an image and measuring the area, integrated density and mean gray value.

Using the calculation for corrected total cell fluorescence as described by McCloy et al.[86]:

$$CTCF = \text{integrated density} - (\text{area of selected cell} \\ \times \text{mean fluorescence of background readings}) \tag{1}$$

For each image, at least five or more background areas were used to normalize against autofluorescence/noise. Data were plotted and analyzed using GraphPad Prism 10 software.

### Mouse strains

The Townes model of SCT (hα/hα::βᴬ/βˢ) was generated by Dr. Tim Townes's laboratory and obtained through Jackson Laboratory (Stock No. 013071)[87]. The Rosa26-Cas9 knockin mouse was generated by Dr. Feng Zhang and purchased from Jackson Laboratory (Stock No. 026179)[88]. The strain was kept in a C57BL/6 J pure background. All animal studies and procedures were approved by the UTMDACC Institutional Animal Care and Use Committee (IACUC) protocol 00001158. Mice were housed under specific pathogen–free conditions with a 12 h light/dark cycle, ambient temperature maintained at 20–24 °C, and relative humidity at 40–60%.

### RMC syngeneic mouse cell line

*MSRT1* (Melinda Soeung renal tumor 1) was generated using CRISPR-Cas9 gene editing technology as previously described[7].

### Genotyping conditions

**Townes model (hα/hα::βᴬ/βˢ).** The common (mouse beta KI) forward primer is 5′- TTGAGCAATGTG GACAGAGAAGG; the Beta A reverse primer is 5′- GTTTAGCCAGGGACCGTTTCAG; the Beta S reverse primer is 5′- AATTCTGGCTTATCGGAGGCAAG; the reverse primer for mouse beta wild type is 5′- ATGTCAGAAGCAAATGTGAGGAGCA. The separated PCR conditions are as follows: 94 °C for 2 min, continue PCR for 10 cycles at 94 °C for 20 s, 65 °C with 0.5 °C decrease per cycle for 15 s, 68 °C for 10 s, continue PCR for 28 cycles at 94 °C for 15 s, 60 °C for 15 s, 72 °C for 10 s.

**Rosa26-Cas9 knockin on C57BL/6 J allele.** The forward primer for wild-type is 5′- CTGGCTTCTGAGGACCG; the reverse primer for wild-type is 5′- AGCCTGCCCAGAAGACTCC. The forward primer for mutant is 5′- GCTAACCATGTTCATGCCTTC; the reverse primer for mutant is 5′- CTCCGTCGTGGTCCTTATAGT. The touchdown PCR cycling conditions are as follows: 10 cycles at 94 °C for 20 s, 65 °C and 0.5 °C per cycle decrease for 15 s, 68 °C for 10 s. Continue the PCR reaction for 28 cycles at 94 °C for 15 s, 60 °C for 15 s, 72 °C for 10 s.

### Immunohistochemistry (IHC)

IHC was performed on FFPE whole kidney sagittal sections. The sagittal kidney sections were fixed in 10% formalin, embedded, and 5 μm sections were cut using a microtome (Leica RM2235). The sections were then baked on slides, de-paraffinized, and treated with citrate buffer (Electron Microscopy Sciences) for antigen retrieval according to the manufacturer's instructions. Endogenous peroxidases were then inactivated using 3% hydrogen peroxide (Sigma-Aldrich) for 10 min followed by washing with phosphate buffer saline. Non-specific signals were then blocked for 20 min using Rodent Block M (Biocare Medical). Samples were then stained with 1:200 primary antibody overnight in 4 °C. After an overnight incubation, slides were washed three times using phosphate buffer saline before staining with Rabbit-on-Rodent HRP polymer (Biocare Medical) for one hour at room temperature. NovaRED peroxidase substrate (Vector Lab, #SK-4800) or 3,3′-Diaminobenzidine (DAB) was used for HRP detection (10 min of exposure). Hematoxylin was used for counterstaining. A Nikon EclipseTi microscope and Nikon DS-Fi1 digital camera were used to capture 20x images. The following primary antibodies were used for IHC analysis:

Phospho-p44/42 MAPK (Erk1/2) (Thr202/Tyr204) (Cell signaling technology, #9101, 1:200); Phospho-MEK1/2 (Ser217/221) (Cell signaling technology, #9121, 1:200); Ki-67 (D2H10) (Cell signaling technology, #9027, 1:200), and GFP (D5.1) (Cell signaling technology, #2956, 1:200).

Equal areas of each 20x image were then quantified using ImageJ/FIJI. Using FIJI, the IHC images were deconvoluted into three channels. The area of the DAB channel images were then quantified using ImageJ/FIJI and graphed into GraphPad for further statistical analysis. All DAB quantifications were normalized to hematoxylin counterstain to account for the number of cells within each image.

## Single-guide RNA design and validation
Single-guide RNAs (sgRNAs) were designed with "GenScript CRISPR sgRNA Design Tool". First, 5'-phosphorylated oligos were annealed and diluted 1:20. Then 1 uL of each annealed and diluted sgRNA was cloned in digested lentiCRISPR V2 (addgene #52961) according to Dr. Feng Zhang's protocol. NEB® Stable Competent *E. coli* (C3040I) colonies resistant to ampicillin antibiotic selection were amplified, and the presence of sgRNA was confirmed by Sanger sequencing. Positive clones were transfected individually in 293 cells along with vectors for lentiviral packaging production, PAX2 (addgene #12260) and PMD2G (addgene #12259). MCT cells were infected by the lentivirus carrying a specific sgRNA and selected for puromycin resistance. The cut efficiency of sgRNA was tested using Western blot analysis to detect protein.

## sgRNA sequences
**sgIFNGR1 (mouse).** AAAAAGAATCTGACTATGCA.

## Surgical procedures
**Orthotopic injection in kidney.** First, $10^{10}$ adeno-associated viral particles were resuspended in PBS (Thermo Fisher Scientific) and Matrigel matrix (Corning) 1:1 solution. Six- to nine-week-old mice were shaved and anesthetized using isoflurane (Henry Schein Animal Health). Analgesia was achieved with buprenorphine SR (0.1 mg/Kg BID) via subcutaneous injection, and shaved skin was disinfected with 70% ethanol and betadine (Dynarex). A 1-cm incision was performed on the left flank through the skin/subcutaneous, and muscular/peritoneal layers. The right kidney was exposed, and 40 µL of cell suspension (500,000 cells) was introduced by Hamilton syringe into the organ by a subcapsular injection. Hemostasis was controlled with a bipolar cautery (Bioseb) if needed. The kidney was carefully repositioned into the abdominal cavity, and the muscular/peritoneal planes were closed individually by absorbable sutures. The skin/subcutaneous planes were closed using metal clips. Mice were monitored daily for the first three days, and twice/week thereafter for signs of tumor growth by manual palpation.

## In vivo treatment study with combination immune checkpoint therapy
Syngeneic mouse model used for in vivo studies were generated by crossing the Townes SCT mouse with the Rosa26-Cas9 knock-in mouse strain that had been kept on a C57BL/6 J pure background[7]. 500,000 cells of the MSRT1 cell line were then injected subcapsular into the right kidneys of mice. Mixed gender cohorts of male and female mice aged 2–4 months were used in an immunotherapy pre-clinical study. Three days after initial subcapsular injection of 500,000 cells of MSRT1 cell line into the right kidneys of mice, mice were treated with either IgG control (*InVivo*Plus polyclonal Syrian hamster IgG, Catalog #BP0087) or a combination of anti-CTLA4 (*InVivoPlus* anti-mouse CTLA-4 (CD152), clone 9H10, #BP0131) and anti-PD1(*InVivoPlus* anti-mouse PD-1 (CD279), clone RMP1-14, #BP0146) via intraperitoneal injection. All treatments were diluted in phosphate buffer saline (PBS) at 5 mg/kg. Mice were treated with 5 mg/kg every three days for a total of eight doses (day 24 was the last day). For studies using IACS 16898,

cohorts of mice were treated simultaneously with a combination of 5 mg/kg of anti-CTLA4 and 5 mg/kg of anti-PD1, or with IgG control twice daily until day 25. IACS 16898 (50 mg/kg) was administered orally via oral gavage with a minimum of 6 h between the first and second doses. Mice were given reprieve from treatment for two days over the weekend. On day 30, magnetic resonance imaging (MRI) was used to detect tumor growth, and mice were euthanized at the end of treatment after undergoing MRI.

## Euthanasia, necropsy, and tissues collection
Mice were euthanized by exposure to $CO_2$ followed by cervical dislocation. A necropsy form was filled in with mouse information, tumor size and weight, infiltrated organ annotations, and metastasis number and location. Mice were euthanized upon detection of physiological distress, paralysis, or other moribund condition resulting in a recommendation of euthanasia by veterinary staff, or when tumors reached 1.5 cm³ in size. This maximal tumor size was not exceeded in this study.

## Magnetic resonance imaging (MRI) Imaging
This procedure has been described in previous studies. A 7 T Bruker Biospec (BrukerBioSpin), equipped with 35 mm inner diameter volume coil and 6 cm inner-diameter gradients, was used for imaging the animals. A fast acquisition with relaxation enhancement (RARE) sequence with TR/TE of 2000/38 ms, matrix size 256 × 192, 0.75 mm slice thickness, 0.25 mm slice gap, 4 × 3 cm FOV, 101 kHz bandwidth, 3 NEX was used for acquired in coronal and axial geometries a multi-slice T2-weighted images. To reduce the respiratory motion, the axial scan sequences were gated.

## Virus preparation
Infectious viral particles were produced using helper plasmids psPAX2 and pMD2G obtained through Addgene. 293 T cells were cultured in DMEM containing 10% FBS (Gibco), 100 µg/ml Penicillin/Streptomycin (Gibco) and transfected using the Polyethylenimine (PEI) method. Virus-containing supernatant was collected 72 h after transfection, spun at 3000 rpm for 10 min and decanted[89]. High-titer preps were obtained by a single round of ultracentrifugation at $>50,000 \times g$ for 2.5 h.

## Histone extraction
To extract histones for protein analysis of H3 K27Ac, the Histone Extraction Kit (ab113476) was used following the manufacturer's protocol.

## IACS-16898 pharmacological characterization studies
**AlphaScreen.** Interactions between the bromodomains of CBP or BRD4 and an acetylated histone H4 peptide were quantified in the presence or absence of small-molecule inhibitors using AlphaScreen technology (Perkin Elmer). Recombinant GST-tagged CBP (residues 1081–1197) and BRD4 (residues 49–170) proteins were purchased from BPS Bioscience, and the biotinylated tetra-acetylated H4(1–21) peptide (acetylated at K5, K8, K12, and K16; AnaSpec, Cat. 64989) served as the substrate.

For the CBP assay, reactions contained 5 nM GST−CBP and 20 nM biotinylated H4 peptide in a total volume of 15 µL assay buffer (50 mM HEPES, pH 7.5; 100 mM NaCl; 1 mM TCEP; 0.003% Tween-20). Test compounds were added at varying concentrations and incubated for 30 min at room temperature. Detection was then initiated by adding 15 µL of buffer (BPS Bioscience, Cat. 33006) containing 7 µg/mL glutathione AlphaLISA acceptor beads (PerkinElmer, AL109) and 14 µg/mL streptavidin donor beads (PerkinElmer, Cat. 676002). Plates were incubated for 2 h in the dark at room temperature, and luminescence was recorded on an EnVision Multilabel Plate Reader (Perkin Elmer). A non-acetylated biotin−H4(1−21) peptide (AnaSpec, Cat. 62555) and 0.25% DMSO were used as negative controls.

For the BRD4 assay, 2.5 nM GST–BRD4 and 10 nM biotinylated H4(1–21) AcK5/8/12/16 peptide were used under the same buffer and detection conditions. Dose–response data were analyzed with Genedata Screener software using a variable-slope fit, treating signal intensity and compound concentration as known parameters.

**BromoMAX and bromoKdELECT.** Screening of IACS-16898 against 32 bromodomain proteins was performed by bromoMAX (DiscoverX). To measure $K_{ds}$, bromoKdELECT was used (DiscoverX).

**In vivo pharmacokinetic properties of IACS-16898 in mouse, rat, dog and monkey.** Male C57BL/6 mice were dosed with IACS-16898 1 mg/kg IV or 10 mg/kg PO, Sprague Dawley (SD) rats were dosed with IACS-16898 1 mg/kg IV or 3 mg/kg PO, male beagle dogs were dosed with IACS-16898 1 mg/kg IV or 3 mg/kg PO, and male cynomolgus monkeys were dosed with IACS-16898 1 mg/kg IV. Plasma samples were harvested pre-dose and at various timepoints after single dose for LC-MS/MS analysis. PK parameters such as clearance rate (Cl), distribution volume (Vdss), terminal half-life through IV dosing (T1/2), and bioavailability (F%) were calculated following the standard formula. $N = 3$ animals per data point. All PK and in vivo experiments were conducted in accordance with the animal welfare procedures and were approved by the UTMDACC IACUC.

**Mouse.** Male C57BL/6 mice (6–8 weeks old, 20–30 g; $n = 9$ per route) were used for intravenous (IV) and oral (PO) dosing studies. Animals had unrestricted access to food and water throughout the experiment. The compound was administered either through the tail vein (IV) or by oral gavage (PO). Blood was collected from each animal at baseline and at 0.083, 0.25, 0.5, 1, 2, 4, 8, and 24 h after dosing into $K_2$EDTA-coated tubes (three animals per time point, with three time points sampled per animal). Plasma was separated by centrifugation at 4 °C and stored at −70 °C until analysis. Drug concentrations were quantified by liquid chromatography–tandem mass spectrometry (LC–MS/MS).

**Rat.** Male Sprague–Dawley rats (6–8 weeks old, 200–300 g; $n = 3$ per route) were used for pharmacokinetic evaluation. Animals were fasted overnight before dosing and refed four hours afterward, with water provided ad libitum. The compound was administered through the dorsal foot vein (IV) or by oral gavage (PO). Blood samples were collected from the tail vein at predose and at the same postdose time points described above. Samples were processed and analyzed by LC–MS/MS as described for mice.

**Dog.** Male Beagle dogs (3–36 months old, 7–10 kg; $n = 3$ per route) received the compound either via the cephalic vein (IV) or by oral gavage (PO). Animals were fasted overnight and allowed food four hours postdose. Blood samples were taken from the saphenous or cephalic vein at predose and at 0.083, 0.25, 0.5, 1, 2, 4, 8, and 24 h following administration. Plasma was isolated and stored at −70 °C before LC–MS/MS quantification.

**Monkey.** Male cynomolgus monkeys (3–5 years old, 3–5 kg; $n = 3$ per route) were used for the nonhuman primate studies. Animals were fasted overnight and fed four hours after dosing. The compound was delivered by cephalic vein injection (IV) or nasogastric gavage (PO). Blood was collected from the saphenous or cephalic vein at baseline and at the same series of postdose time points used for other species. Plasma was separated by centrifugation at 4 °C, stored at −70 °C, and analyzed using LC–MS/MS.

**In vivo PK profiling of IACS-16898 in mice.** To calculate PK parameters, SCID mice were dosed with IACS-16898 via oral administration at 50 mg/Kg, BID, for 5 days. Plasma samples were collected at indicated times (0, 0.5, 1, 2, 8 and 16 h) post dose for LC-MS/MS analysis.

PK parameters were calculated following standard formulas. $N = 3$ animals per data point.

**LC-MS/MS quantification of IACS-16898 in mouse plasma.** Concentrations of IACS-16898 in mouse plasma were quantitated using a validated LC-MS/MS method. For each analysis, 25 μL of plasma sample was precipitated with 200 μL of acetonitrile containing imipramine (70 ng/mL) as an internal standard. This suspension was vortexed for 10 min and centrifuged on a benchtop centrifuge at $1800 \times g$ for 10 min, after which 100 μL of the supernatant was transferred to a new tube and diluted with 200 μL of water prior to LC-MS/MS analysis. Plasma samples with concentrations above ULOQ (Upper Limit of Quantitation) were diluted using the blank mouse plasma as necessary. LC-MS/MS analysis of IACS-16898 was conducted on a Shimadzu Nexara UHPLC system coupled with a Sciex 5500 triple quadrupole mass spectrometer operated at the positive mode (ESI +). The detection conditions of the mass spectrometer were as follows: gas1, 35 psi; gas2, 50 psi; curtain gas, 35 psi; source temperature, 400 °C; and ion spray voltage, 5500 V. IACS-16898 was separated using a Supelco Ascentrix C18 column (1.7 μm, 2.1 mm × 20 mm) and detected by a multiple reaction monitoring transition (m/z 539.2 > 482.2 for IACS-16898 and 281.1 > 86.1 for imipramine). The injection volume was 2 μL. The LC mobile phase A was 0.1% formic acid-water and B was 0.1% formic acid-acetonitrile. The gradient was 5% B (0-0.3 min), 5–95% B (0.3–1.3 min), 95% B (1.3 to 1.7 min), 95-5% B (1.7–1.71 min), 5% B (1.71 to 2.3 min), and the flow rate was 0.5 mL/min. The column temperature was 40 °C. Under these conditions, the retention time was 1.10 min for IACS-16898 and 1.13 min for the internal standard. The method was validated with the analytical range of 1 – 1000 ng/mL IACS-16898 in untreated mouse (SCID) plasma.

**Proliferation assay.** DOHH2 cells were kept in RPMI Medium 1640 (Gibco, Cat# 11875-119) supplemented with 10% Fetal Bovine Serum (Sigma, Cat# F2442), at 37 °C, 5% $CO_2$ in a humidified incubator. $2.5 \times 10^3$ cells were seeded in 96-well plates for seven days. Treatments were performed in triplicate with IACS-16898 diluted in DMSO vehicle at indicated concentrations, starting from 10 μM, followed by a serial dilution of 1:4 to a final concentration of 0.00061 μM, or DMSO only as a control, in a final volume of 100 μl. Proliferation was measured using the Cell Titer GLO assay (Promega), according to the manufacturer's instructions. Data were collected, replicates were averaged, standard deviations calculated and analyzed through nonlinear regression analysis with 4 parameters to generate $IC_{50}$ values.

**Western blot analysis for histones.** Histones were purified using the Histone extraction kit from Abcam (Ab113476), according to the manufacturer's instructions. Briefly, cells were collected and washed them twice with cold PBS. 100 μl of pre-lysis buffer were added and incubated on ice for 10 min. Samples were centrifuged at $10,000 \times g$ for 1 min at 4 °C. Pellets were then resuspended in 90 μl of lysis buffer and incubate on ice for 30 min. Samples were then centrifuged at $15,000 \times g$ for 5 min at 4 °C, and the supernatant fraction transferred into a new vial. 0.3 volumes of Balance-DTT buffer were added to the supernatant immediately. 5 μg of histone extracts were prepared using standard methods, separated by SDS-PAGE gel electrophoresis, followed by western blot detection. Total H3 antibody (Cell Signaling Technologies, #4499) and Acetyl-H3K27 antibody (Cell Signaling Technologies, #8173) were used for probing and chemiluminescence was used for signal detection. Uncropped scanned films are provided in Supplementary Figs. S36, S37.

**In vivo testing of IACS-16898.** All in vivo work was approved by the UTMDACC IACUC under protocol # 00000884. 6–8-week-old female CB-17 SCID mice (Taconic) were injected subcutaneously with $1 \times 10^6$ DOHH2 cells, resuspended in 1:1 Matrigel mixture. Tumors were

allowed to reach 200 mm³ on average, then the mice were randomized in treatment cohorts of 7 and treated with 0.5% methylcellulose as a vehicle control or 50 mg/Kg of IACS-16898 twice per day, on a 5 on/ 2 off schedule. Tumor volume and body weight were measured. Data were analyzed using GraphPad Prism 10. Day 0 was the day before treatment started. Body weight (BW) change, and tumor growth inhibition (TGI) were calculated based on the following formulas:

$$Body\ Weight\ Change\% = (BW_i - BW_0)/BW_0 * 100\% \qquad (2)$$

BW_i and BW_0 are body weight of an individual mouse on measurement day i and on day 0, respectively.

$$TGI\% = 100 - ([V_{end} - trx - V_{start} - trx/V_{end} - veh - V_{start} - veh]*100) \qquad (3)$$

(trx = treatment; veh = vehicle; V_{end} = final volume; V_{start} = initial volume).

Tumor size was measured with caliper and calculated using a standard formula: length × width²/2. Euthanasia was completed by continued exposure to $CO_2$ followed by cervical dislocation to ensure death according to approved protocol by MDACC IACUC.

**RNA extraction and real-time PCR analysis of DOHH2 samples treated with IACS-16898.** Subcutaneous tumors were harvested and homogenized with stainless-steel bead-beating homogenizer. RNA was isolated using the RNeasy mini-RNA isolation kit (Qiagen #74106) according to the manufacturer's instructions. Quantitative real-time PCR was conducted using the following Taqman probes and Taqman universal master mix II without UNG by Thermo Fisher: BCL6, Assay ID: Hs00153368_m1; endogenous control, human RPLP0 (Thermo Fisher #4326314E). The ΔΔCt method was used to analyze the qRT-PCR data. Statistical analysis was performed using GraphPad Prism 10, by paired $t$ test, and changes were considered significant for $P < 0.05$.

**Reporting summary**
Further information on research design is available in the Nature Portfolio Reporting Summary linked to this article.

## Data availability
The trial protocol is available with this submission as a Supplementary Note in the Supplementary Information file. All processed sequencing data (scRNA-seq and bulk RNA-seq) generated in this study have been deposited in the NCBI Gene Expression Omnibus (GEO) under accession code GSE256326. Raw sequencing data are protected and are not available due to data privacy laws and a lack of patient consent. Requests to access raw data should be forwarded to the corresponding authors at PMsaouel@mdanderson.org and/or ggenovese@mdanderson.org and/or jgao1@mdanderson.org and/or lwang22@mdanderson.org. All requests for data and materials will be promptly reviewed to verify whether the request is subject to any intellectual property or confidentiality obligations. Any data and materials that can be shared will be released via a Material Transfer Agreement. The remaining data are available within the Article, Supplementary Information or Source Data file. Source data are provided in this paper.

## Code availability
Codes for analysis of scRNA-seq and bulk RNA-seq data, as well as for generating Fig. 2 and Supplementary Figs. S2–S8, S15–S18 are available at GitHub (https://github.com/XinmiaoYan/RMC_hyperprogression) with the version used in this manuscript linked to Zenodo (https://doi.org/10.5281/zenodo.16995867)[90].

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

## Acknowledgements

We thank all of the patients and their families who participated in this phase 2 study. This is an investigator-sponsored trial developed and conducted at The University of Texas MD Anderson Cancer Center. Bristol Myers Squibb supplied funding, nivolumab and ipilimumab for the clinical trial and had no role in study design, conduct, data collection, data analysis or the writing of this report. This work was supported by the National Cancer Institute through the Cancer Center Support Grant P30CA16672 (Institutional Tissue Bank (ITB) and Research Histology Core Laboratory (RHCL)) and supported by the generous philanthropic contributions to The University of Texas MD Anderson Cancer Center Moon Shots Program™, Adaptive Patient-Oriented Longitudinal Learning and Optimization (APOLLO) Moonshot Program, and the Translational Molecular Pathology-Immunoprofiling (TMP-IL) Platform at the Department Translational Molecular Pathology, the University of Texas MD Anderson Cancer Center. The clinical and preclinical studies were additionally supported in part by MD Anderson's Prometheus informatics system and the Department of Genitourinary Medical Oncology's Eckstein Laboratory. Pre-sequencing processing work was carried out by the Moon Shots Platform CGL, The University of Texas MD Anderson Cancer Center, Sheikh Khalifa Bin Zayed Al Nahyan Institute for Personalized Cancer Therapy. Sequencing and data generation was supported by CA016672(ATGC) grant from The University of Texas MD Anderson Cancer Center, Advanced Technology Genomics Core. The investigator-initiated clinical trial was additionally supported in part by Bristol Myers Squibb. P Msaouel is supported by the National Cancer Institute R37CA288448 and R01CA285454, the Andrew Sabin Family Foundation Fellowship, Gateway for Cancer Research, a Translational Research Partnership Award (KC200096P1) and an Idea Development Award (RA230062) by the United States Department of Defense, an Advanced Discovery Award by the Kidney Cancer Association, a Translational Research Award by the V Foundation, the MD Anderson Physician-Scientist Award, donations from the Renal Medullary Carcinoma Research Foundation in honor of Ryse Williams, as well as philanthropic donations by the Chris "CJ" Johnson Foundation, and by the family of Mike and Mary Allen. L. Perelli was supported by the Cancer Prevention and Research Institute of Texas (RR250083), a Department of Defense Discovery Award (PR240469) and a National Institute of Neurological Disorders and Stroke R03 Award (R03NS145168). G. Genovese was supported by the National Cancer Institute R01CA258226. Development of IACS-16898 was supported by The Mark Foundation for Cancer Research.

## Author contributions

All authors collected and analyzed the data and provided a critical review of the manuscript. P.M., N.M.T., and R.S.T. designed the clinical study. P.M., N.M.T., R.S.T., T.K.B., P.R., and N.C.D. monitored and

interpreted clinical data. P.M., J.G., L.W., and X.Y. designed the translational correlative studies and interpreted their results. P.M., M.S., and G.Ge (Giannicola Genovese). designed the preclinical experiments and interpreted their results. M.S., P.M., C.Z., J.Q., M.K., F.D., H.K., L.Z., D.H.P., M. W., R.H., Z.C., L.P., C.L., T.N.A.L., N.B., R.S., I.L.H., J.G., C.C.C., N.F., K.L., G.Ga (Guang Gao)., T.L.P., F.M., Y.J., Q.A.X., N.M.Z., D.N.T., C.L.W., M.M.M., J.Z., G.H., Y.C., R.W., M.D., E.D., F.P., Y.L., A.F., A.V., M.J.S., P.J., J.R.M., T.H., and G.F.D. supported and carried out preclinical studies and contributed to data interpretation. J.C., P.K.C., R.A.S., A.J., W.L., C.A.A., L.C.C., E.R.P., H.L., C.L.H., and I.I.W., and A.F. contributed to the collection and analysis of the translational correlative studies. P.M. and N.M.T. enrolled patients on the study. R.S.T. and X.Y. contributed to the statistical analysis. Development of the first draft of the manuscript was led by P.M., M.S., X.Y., J.G., L.W., and G.Ge. All authors contributed to drafting the manuscript and provided final approval. P.M., J.G., L.W., and G.Ge. had final responsibility for the decision to submit for publication.

## Competing interests

P. Msaouel has received honoraria for service on a Scientific Advisory Board for Mirati Therapeutics, Bristol Myers Squibb, and Exelixis; consulting for Axiom Healthcare Strategies; non-branded educational programs supported by DAVA Oncology, Exelixis and Pfizer; and research funding for clinical trials from Regeneron Pharmaceuticals, Summit Therapeutics, Merck, Takeda, Bristol Myers Squibb, Mirati Therapeutics, Gateway for Cancer Research, and UT MD Anderson Cancer Center. J. Gao serves as a consultant for AstraZeneca, Aveo Pharmaceuticals, CRISPR Therapeutics, Infinity Pharmaceuticals, Janssen, Jounce, Pfizer, Polaris, and Symphogen. N. M. Tannir reported receiving and personal fees (honoraria) from Calithera Biosciences during the conduct of the study; and grants (sponsored trial) from Calithera Biosciences, Bristol Myers Squibb (BMS), Nektar Therapeutics, Arrowhead Pharmaceuticals, and Novartis, as well as personal fees (honoraria) from Calithera Biosciences, BMS, Eisai Medical Research, Merck Sharp & Dohme (MSD), Deka Biosciences, Neoleukin Therapeutics, Exelixis, and Ono Pharmaceutical outside the submitted work. The remaining authors declare no competing interests.

## Additional information

Melinda Soeung [1,19], Xinmiao Yan[1,19], Ciro Zanca[2], Jing Qian[3,4], Menuka Karki[3,4], Fei Duan[3,4], Hania Khan[3], Li Zhang [3], David H. Peng[2], Mariah Williams[2], Rong He[3,4], Ziheng Chen [1], Luigi Perelli [3,5], Jianfeng Chen[3,4], Rebecca S. Tidwell [6], Pankaj K. Chauhan [3,4], Courtney N. Le [3], Truong N. A. Lam[3], Nirjar Bhattacharya[1], Rutvi Shah [1,7], I-Lin Ho[1], Jason P. Gay[2], Caroline C. Carrillo[2], Ningping Feng[2], Kang Le[8], Guang Gao[2], Teresa L. Perry[8], Faika Mseeh[8], Yongying Jiang[8], Quanyun A. Xu[8], Niki Marie Zacharias [9], Rahul A. Sheth [10], Tharakeswara K. Bathala[11], Priya Rao [12], Najat C. Daw[13], Durga N. Tripathi[14], Cheryl L. Walker[14], Mohammad M. Mohammad[15], Jianhua Zhang[1], Guangchun Han [1], Yanshuo Chu[1], Ruiping Wang[1], Minghao Dang [1], Enyu Dai[1], Fuduan Peng[1], Yunhe Liu [1], Akshaya Jadhav [16], Wenhua Lang[16], Claudio A. Arrechedera [16], Leticia Campos Clemente [16], Edwin R. Parra [16], Hsinyi Lu[16], Cara L. Haymaker [16], Ignacio I. Wistuba[16], Andrew Futreal [1], Andrea Viale[1], Michael J. Soth[8], Philip Jones [8], Joseph R. Marszalek [8], Timothy Heffernan [2], Giulio F. Draetta [2], Nizar M. Tannir [3], Jianjun Gao [3,4,20] ✉, Linghua Wang [1,7,17,18,20] ✉, Giannicola Genovese [1,2,3,4,20] ✉ & Pavlos Msaouel [3,4,7,16,20] ✉

[1]Department of Genomic Medicine, The University of Texas MD Anderson Cancer Center, Houston, TX, USA. [2]Translational Research to Advance Therapeutics and Innovation in Oncology (TRACTION), The University of Texas MD Anderson Cancer Center, Houston, TX, USA. [3]Department of Genitourinary Medical Oncology, The University of Texas MD Anderson Cancer Center, Houston, TX, USA. [4]David H. Koch Center for Applied Research of Genitourinary Cancers, The University of Texas, MD Anderson Cancer Center, Houston, TX, USA. [5]Department of Cancer Biology, The University of Texas, MD Anderson Cancer Center, Houston, TX, USA. [6]Department of Biostatistics, The University of Texas MD Anderson Cancer Center, Houston, TX, USA. [7]The University of Texas MD Anderson Cancer Center UTHealth Houston Graduate School of Biomedical Sciences (GSBS), Houston, TX, USA. [8]Institute for Applied Cancer Science (IACS), Therapeutics Discovery Division, The University of Texas MD Anderson Cancer Center, Houston, Texas, USA. [9]Department of Urology, The University of Texas MD Anderson Cancer Center, Houston, TX, USA. [10]Department of Interventional Radiology, The University of Texas MD Anderson Cancer Center, Houston, TX, USA. [11]Department of Diagnostic Imaging, The University of Texas MD Anderson Cancer Center, Houston, TX, USA. [12]Department of Pathology, The University of Texas MD Anderson Cancer Center, Houston, TX, USA. [13]Department of Pediatrics, The University of Texas MD Anderson Cancer Center, Houston, TX, USA. [14]Center for Precision Environmental Health, Baylor college of Medicine, Houston, TX, USA. [15]Institute for Personalized Cancer Therapy, The

University of Texas MD Anderson Cancer Center, Houston, TX, USA. [16]Department of Translational Molecular Pathology, The University of Texas MD Anderson Cancer Center, Houston, TX, USA. [17]The James P. Allison Institute, The University of Texas MD Anderson Cancer Center, Houston, TX, USA. [18]Institute for Data Science in Oncology, The University of Texas MD Anderson Cancer Center, Houston, TX, USA. [19]These authors contributed equally: Melinda Soeung, Xinmiao Yan. [26]These authors jointly supervised this work: Pavlos Msaouel, Giannicola Genovese, Linghua Wang, Jianjun Gao.
✉e-mail: jgao1@mdanderson.org; lwang22@mdanderson.org; ggenovese@mdanderson.org; pmsaouel@mdanderson.org

