## [Transparent Peer Review file · Nature Communications]

Nivolumab plus ipilimumab induces hyperprogression in renal medullary carcinoma: results of a phase II trial and preclinical evidence

Corresponding Author: Dr Pavlos Msaouel

Version 0:

Reviewer comments:

Reviewer #1

(Remarks to the Author)

The authors has adressed my comments (reviewer #1).

Reviewer #4

(Remarks to the Author)

The authors report findings from a clinical trial concerning renal medullary carcinoma. Due to its extreme aggressiveness and its incidence in younger patients, the study is very relevant and needed.

The authors illustrate the importance of P300 in IFNGR1KO cells, of which inhibition resulted in reduced tumour growth in mouse models. The paper is very clear to read and figures are presented in an attractive manner, but its single-cell analysis suffers from low generalisation potential due to the very small patient cohort (7 samples in 3 locations) and absence of paired information. While this is typical of RMC studies, the low power of the cohort in combination with the composition of the single-cell object (tissue sampling site heterogeneity and treatment states) are highly problematic. Despite the issues, the study represents one of the larger resources associated with RMC and this should be taken into account.

Comments:

- The paper orients its P300 marker discovery based on single-cell analysis. Due to the diversity in sampling sites, all treated samples are in the liver (Supplementary S3). Meanwhile, baseline samples cover lymph node, primary kidney and one liver sample. As such, comparing treated versus untreated is incorrect as the reference has a very different composition. The analysis should be tissue-specific, yet then the contrast is based on a single unpaired liver sample. As such, in the correct setting, no distinction between patient-specific or conserved patterns can be made. In the latter liver-specific case, the title should be adjusted to reflect validity only in liver metastases. Simultaneously, the number of cells studied is extremely low in the correct comparison, which will open issues for dropouts in single-cell analysis (liver: 319 baseline vs 634 post). The authors did perform the tissue-specific analysis (see supplementary), but while S100A9 is a consistent hit, EP300 does not appear in the marked genes. While its p-value is significant, it is very low for single-cell differential gene expression analysis ($p=0.0049$) and the question remains if the effect size is meaningfully different. While the study limitations are declared in the text, it remains problematic to use the scRNA to guide the rest of the study.

- S100A9 is highly altered in the dataset. However, this is a marker that is expressed in a wide range of cell types, as demonstrated by the multitude of single-cell atlases published in the past few years. While the authors do show higher plasticity, it remains unclear if the shift is really towards macrophage mimicry. The authors use Euclidean and Manhattan distance to describe transcriptional similarity. Single-cell analysis is inherently highly dimensional, especially if the authors decide to use the top 3000 genes. As such, the distance metric aggregates noise from a large number of dimensions. To estimate transcriptional similarity across the expression vector, it is often more helpful to look at directional similarity. Furthermore, Euclidean distance suffers from sparsity issues and is not scale-invariant. A study that discusses the impact of

distance metrics can be found here: <https://academic.oup.com/bib/article/23/6/bbac387/6712300>. As such, I feel this analysis is currently insufficient to argue the trend of myeloid mimicry in humans as compared to typical cancer-associated changes in cell cycling.

- Figure 2m should reflect the situation in liver metastases to be representative of treatment-induced changes (in the liver metastases), not the tissue-specific alterations-treatment state confounding effect.

- The current manuscript feels forced towards myeloid mimicry. Have the authors looked at changes in the tumour micro-environment and cell-to-cell communication? It could be interesting to run tools, such as CellChat or CellPhoneDB and characterize the different interactions in the liver metastatic setting.

- The authors show that the IFNGR1 knockout has lower p300 content and that these cells were more susceptible to ICT. Simultaneously, in vitro tests on MSRT1 cells show increased F4/80 levels with increasing IFN exposure, which is negated by IFNGR1KO. However, there is no indication of transcriptional representativeness of these cell lines with regards to the primary tumour or the liver metastases in human. In addition to this, F4/80 expression can vary depending on antibody clone used. It would be beneficial to add other markers to complement F4/80, such as CD11b, CD11c and Ly6C.

Reviewer #5

(Remarks to the Author)

This is a phase 2 open label single arm clinical trial testing the efficacy of combination nivolumab plus ipilimumab in patients with locally advanced or metastatic RMC. The primary endpoint is ORR. A planned sample size is 30 with safety and futility monitoring after every 10 patients. With only one patient showing an unconfirmed PR, the trial was halted for futility after enrolling 10 patients. The design, analysis and results are well described.

The sample size justification used Bayesian posterior probability. The prior of ORR is beta(0.6, 1.4) which is a skeptical prior that centered around historical rate of 0.29 with some variation. The reference is also from beta distribution instead of using point estimate to represent uncertainty. Then a posterior comparison of two independent beta distribution is done and a posterior probability threshold is prespecified. line 470-472 "Point values such as the historical response rate of 29% can be mapped to a posterior distribution of the values given the data. If the point value, in this case 0.29, lies at the extremes of the posterior distribution then the results are less compatible with this value" is confusing and need rewording.

For the futility monitoring, the prior of ORR is beta (0.5, 0.5) which is non-informative prior that places higher density on extreme rates. The trial will stop if posterior probability of $ORR < 0.3$ is higher than 0.9. The goal is to detect early if the treatment is unlikely to succeed, this prior is more aggressive futility stopping if the early data is disappointing. A sensitivity analysis is recommended to compare with Beta(0.6, 1.4) prior.

Version 1:

Reviewer comments:

Reviewer #4

(Remarks to the Author)

In their manuscript, the authors describe renal medullary carcinoma (RMC), which is a rare malignancy. The authors provide a thoroughly revised manuscript, in which disclaimers on dataset limitations are clearly described (e.g. "However, the lack of matched

baseline and post-treatment samples for single-cell transcriptomics precluded definitive conclusions." or "they remain murine models and do not entirely mirror the transcriptomic complexity of human RMC"). Titles have also been adjusted for increased clarity and the authors have taken substantial steps in validating stability of their single-cell RNA-seq signals.

The authors have satisfactorily addressed all my previous concerns and I would like to congratulate them with their revision efforts.

Reviewer #5

(Remarks to the Author)

Thanks for response and updates, I have no concerns.

Point-by-point response

Reviewer #4:

The authors report findings from a clinical trial concerning renal medullary carcinoma. Due to its extreme aggressiveness and its incidence in younger patients, the study is very relevant and needed.

The authors illustrate the importance of P300 in IFNGR1KO cells, of which inhibition resulted in reduced tumour growth in mouse models. The paper is very clear to read and figures are presented in an attractive manner, but its single-cell analysis suffers from low generalisation potential due to the very small patient cohort (7 samples in 3 locations) and absence of paired information. While this is typical of RMC studies, the low power of the cohort in combination with the composition of the single-cell object (tissue sampling site heterogeneity and treatment states) are highly problematic. Despite the issues, the study represents one of the larger resources associated with RMC and this should be taken into account.

***Reviewer:** The paper orients its P300 marker discovery based on single-cell analysis. Due to the diversity in sampling sites, all treated samples are in the liver (Supplementary S3). Meanwhile, baseline samples cover lymph node, primary kidney and one liver sample. As such, comparing treated versus untreated is incorrect as the reference has a very different composition. The analysis should be tissue-specific, yet then the contrast is based on a single unpaired liver sample. As such, in the correct setting, no distinction between patient-specific or conserved patterns can be made. In the latter liver-specific case, the title should be adjusted to reflect validity only in liver metastases.*

***Authors:** We thank the reviewer for the thoughtful critique. The corresponding Results subsection titles have been changed to explicitly read: “Myeloid-like transcriptional features emerge in **RMC liver metastases** after treatment with nivolumab plus ipilimumab” and “**RMC liver metastases** have increased transcriptional similarity to myeloid cells after treatment with nivolumab plus ipilimumab”. Furthermore, as detailed in the replies below we have prioritized the scRNA-seq comparison of the liver metastases and performed additional analyses using six complementary metrics (Euclidean, Manhattan, Bray–Curtis and Phi dissimilarities, plus Pearson- and Cosine-based similarities) all of which showed a significantly closer tumor cell distance to myeloid cells in the post-immunotherapy samples.*

***Reviewer:** Simultaneously, the number of cells studied is extremely low in the correct comparison, which will open issues for dropouts in single-cell analysis (liver: 319 baseline vs 634 post). The authors did perform the tissue-specific analysis (see supplementary), but while S100A9 is a consistent hit, EP300 does not appear in the marked genes. While its p-value is significant, it is very low for single-cell differential gene expression analysis ($p=0.0049$) and the question remains if the effect size is meaningfully different. While the study limitations are declared in the text, it remains problematic to use the scRNA to guide the rest of the study.*

***Authors:** Restricting the comparison to the liver metastases did indeed attenuate the EP300 signal. However, its key partner CEBPB remained strongly up-regulated ($p = 2.6 \times 10^{-10}$) as shown in the revised **Supplementary Figure S8**, supporting the biological relevance of the EP300–CEBPB axis even in this more stringent setting. We have accordingly revised the results section to clarify these points.*

Reviewer: *S100A9 is highly altered in the dataset. However, this is a marker that is expressed in a wide range of cell types, as demonstrated by the multitude of single-cell atlases published in the past few years. While the authors do show higher plasticity, it remains unclear if the shift is really towards macrophage mimicry. The authors use Euclidean and Manhattan distance to describe transcriptional similarity. Single-cell analysis is inherently highly dimensional, especially if the authors decide to use the top 3000 genes. As such, the distance metric aggregates noise from a large number of dimensions. To estimate transcriptional similarity across the expression vector, it is often more helpful to look at directional similarity. Furthermore, Euclidean distance suffers from sparsity issues and is not scale-invariant. A study that discusses the impact of distance metrics can be found here: <https://academic.oup.com/bib/article/23/6/bbac387/6712300>. As such, I feel this analysis is currently insufficient to argue the trend of myeloid mimicry in humans as compared to typical cancer-associated changes in cell cycling.*

Authors: We appreciate the reviewer’s careful scrutiny of our similarity analysis and the pointer to Watson *et al.* (Brief. Bioinform. 2022; PMID: 36151725). In the revised manuscript we have substantially expanded and stress-tested this component by recalculating distances with Euclidean, Manhattan, Bray–Curtis and Φ (phi) dissimilarities plus the two orientation-aware metrics (cosine distance and Pearson correlation) the Watson *et al.* article suggested as consistently ranked among the top five metrics with the least sensitivity to sparsity for all structural conditions. All six independently showed a significant decrease in tumor-to-myeloid distance (**RFigure 1** corresponding to the revised **Main Figure 2m-n**), confirming the robustness of our findings.

RFigure 1. (a) Violin plot scRNA-seq analyses showing the distance/dissimilarity (including Euclidean distance, Manhattan distance, Phi and Bray Curtis) and similarity (including cosine and pearson correlation) between liver tumor cells and myeloid cells at baseline and post nivolumab plus ipilimumab (PostNI). **(b)** Bar plot shows the log fold change of the distance/dissimilarity (including Euclidean distance, Manhattan distance, Phi and Bray Curtis) and the change of similarity (including cosine and pearson correlation) between liver tumor cells and myeloid cells at baseline and PostNI.

Reviewer: *Figure 2m should reflect the situation in liver metastases to be representative of treatment-induced changes (in the liver metastases), not the tissue-specific alterations-treatment state confounding effect.*

Authors: We have accordingly revised **Figure 2m-n (RFigure 1** above) to show the liver-restricted analyses.

Reviewer: The current manuscript feels forced towards myeloid mimicry. Have the authors looked at changes in the tumour micro-environment and cell-to-cell communication? It could be interesting to run tools, such as CellChat or CellPhoneDB and characterize the different interactions in the liver metastatic setting.

Authors: We have accordingly now added a dedicated CellChat analysis (RFigure 2 corresponding to the revised **Supplementary Figure S19**) focused on the liver metastasis samples. We found an increase in tumor and myeloid cell interactions in the liver metastases post nivolumab plus ipilimumab compared with baseline. These hypothesis-generating findings suggest that ICT not only selects for a tumor population with myeloid-like transcriptional features but may also rewire paracrine circuits that reinforce macrophage–tumor crosstalk.

RFigure 2. Differential cell-cell communications between PostNI and baseline in liver metastasis. **(a)** Bar plot shows the number of interactions and interaction strength of baseline and PostNI. **(b)** Circle plot shows the differential number of interactions and differential interaction strength across immune and tumor cells between PostNI and baseline. The blue line represents a decreased number or strength, and the red line represents an increased number or strength. **(c)** Heatmap shows the differential number of interactions and differential interaction strength from source cells to target cells between PostNI and baseline. **(d)** The chord plot shows the overall interactions between immune cells as source cells and malignant tumor cells as target cells. The color of the crosstalk represents the source cell color. **(e)** Dotplot shows the specific differential ligand-receptor of interactions of immune cell and malignant cell between PostNI and baseline.

Reviewer: The authors show that the *IFNGR1* knockout has lower p300 content and that these cells were more susceptible to ICT. Simultaneously, *in vitro* tests on MSRT1 cells show increased F4/80 levels with increasing *IFN* γ exposure, which is negated by *IFNGR1*KO. However, there is no indication of transcriptional representativeness of these cell lines with regards to the primary tumour or the liver metastases in human. In addition to this, F4/80 expression can vary depending on antibody clone used. It would be beneficial to add other markers to complement F4/80, such as CD11b, CD11c and Ly6C.

Authors: We have now revised the limitations section in the discussion to note that our murine MSRT1 system may not fully capture the transcriptional landscape of human RMC liver metastases. Furthermore, we evaluated the recommended additional mouse myeloid lineage markers and confirmed that Ly6c and CD11c showed a similar *IFN* γ -dependent pattern as F4/80, whereas CD11b did not show upregulation relative to LPS (**RFigure 3** corresponding to the revised **Supplementary Figure S24**).

RFigure 3. Upregulation of mouse myeloid lineage markers in following in MSRT1 treated with *IFN* γ . **(a-c)** Representative immunofluorescent images of Ly6c **(a)**, CD11b **(b)**, and CD11c **(c)** in MSRT1 cells treated with interferon gamma (*IFN* γ) 100 ng/mL or lipopolysaccharides (LPS) 100 ng/mL *in vitro*. Untreated RAW264.7, a mouse macrophage-like cell line, was used as a positive control. **(d-e)** Quantification of Ly6c **(d)**, CD11b **(d)**, and CD11c **(e)** immunofluorescence in MSRT1 cells treated with *IFN* γ 100 ng/mL, lipopolysaccharides (LPS) 100 ng/mL, and untreated for 48 hours. Data are expressed as mean value \pm SEM, with *P* value calculated by student's *t* test, and graphing was done using Prism.

Reviewer #5:

This is a phase 2 open label single arm clinical trial testing the efficacy of combination nivolumab plus ipilimumab in patients with locally advanced or metastatic RMC. The primary endpoint is ORR. A planned sample size is 30 with safety and futility monitoring after every 10 patients. With only one patient showing an unconfirmed PR, the trial was halted for futility after enrolling 10 patients. The design, analysis and results are well described.

Reviewer: The sample size justification used Bayesian posterior probability. The prior of ORR is $\text{beta}(0.6, 1.4)$ which is a skeptical prior that centered around historical rate of 0.29 with some variation. The reference is also from beta distribution instead of using point estimate to represent uncertainty. Then a posterior comparison of two independent beta distribution is done and a posterior probability threshold is prespecified. line 470-472 “Point values such as the historical response rate of 29% can be mapped to a posterior distribution of the values given the data. If the point value, in this case 0.29, lies at the extremes of the posterior distribution then the results are less compatible with this value” is confusing and need rewording.

Authors: Thank you for this helpful feedback. We agree that these sentences are confusing and have been accordingly revised as follows: “One hypothesis test was planned to compare the historical response rate of 29% (13/45) to the posterior distribution of this trial’s response rate. This would be tested using the posterior probability that the beta-binomial posterior distribution from this trial was greater than the beta-binomial distribution from the historical trial. This hypothesis was only planned to be tested if the trial successfully accrued to 30 patients.”

Reviewer: For the futility monitoring, the prior of ORR is $\text{beta}(0.5, 0.5)$ which is non-informative prior that places higher density on extreme rates. The trial will stop if posterior probability of $\text{ORR} < 0.3$ is higher than 0.9. The goal is to detect early if the treatment is unlikely to succeed, this prior is more aggressive futility stopping if the early data is disappointing. A sensitivity analysis is recommended to compare with $\text{Beta}(0.6, 1.4)$ prior.

Authors: Thank you for identifying a potential issue with the prior and overall design. We have rerun the design with the $\text{Beta}(0.6, 1.4)$ prior as requested, and this resulted in identical stopping rules that would therefore not change the trial outcomes. We have noted this post-hoc sensitivity analysis in the revised Methods.